# Moonlighting chaperone activity of the enzyme PqsE contributes to RhlR-controlled virulence of *Pseudomonas aeruginosa*

Sebastian Roman Borgert[1,8], Steffi Henke [1,8], Florian Witzgall[1], Stefan Schmelz[1], Susanne zur Lage [1], Sven-Kevin Hotop[2], Steffi Stephen[2], Dennis Lübken[3], Jonas Krüger[4], Nicolas Oswaldo Gomez [4], Marco van Ham [5], Lothar Jänsch[5], Markus Kalesse[3], Andreas Pich[6], Mark Brönstrup [2], Susanne Häussler [4] & Wulf Blankenfeldt [1,7] ✉

*Pseudomonas aeruginosa* is a major cause of nosocomial infections and also leads to severe exacerbations in cystic fibrosis or chronic obstructive pulmonary disease. Three intertwined quorum sensing systems control virulence of *P. aeruginosa*, with the *rhl* circuit playing the leading role in late and chronic infections. The majority of traits controlled by *rhl* transcription factor RhlR depend on PqsE, a dispensable thioesterase in *Pseudomonas* Quinolone Signal (PQS) biosynthesis that interferes with RhlR through an enigmatic mechanism likely involving direct interaction of both proteins. Here we show that PqsE and RhlR form a 2:2 protein complex that, together with RhlR agonist *N*-butanoyl-L-homoserine lactone (C4-HSL), solubilizes RhlR and thereby renders the otherwise insoluble transcription factor active. We determine crystal structures of the complex and identify residues essential for the interaction. To corroborate the chaperone-like activity of PqsE, we design stability-optimized variants of RhlR that bypass the need for C4-HSL and PqsE in activating PqsE/RhlR-controlled processes of *P. aeruginosa*. Together, our data provide insight into the unique regulatory role of PqsE and lay groundwork for developing new *P. aeruginosa*-specific pharmaceuticals.

The gram-negative bacterium *Pseudomonas aeruginosa* is infamous for causing hospital-acquired infections and for persisting in the lungs of people with chronic obstructive pulmonary disease or cystic fibrosis, leading to high morbidity and mortality[1]. Due to high intrinsic resistance, treatment options for *P. aeruginosa* are limited, and the emergence of antimicrobial resistance (AMR) further aggravates this shortcoming. In fact, *P. aeruginosa* has recently been identified as one of the leading causes of AMR-related death[2], and the World Health Organization lists carbapenem-resistant *P. aeruginosa* as one of the pathogens for which new treatments are most critically required[3].

*P. aeruginosa* produces a large number of virulence factors that damage tissue and help colonization and survival within the host[4].

[1]Department Structure and Function of Proteins, Helmholtz Centre for Infection Research, Inhoffenstr. 7, 38124 Braunschweig, Germany. [2]Department Chemical Biology, Helmholtz Centre for Infection Research, Inhoffenstr. 7, 38124 Braunschweig, Germany. [3]Institute of Organic Chemistry, Leibniz University Hannover, Schneiderberg 1B, 30167 Hannover, Germany. [4]Department Molecular Bacteriology, Helmholtz Centre for Infection Research, Inhoffenstr. 7, 38124 Braunschweig, Germany. [5]Cellular Proteomics, Helmholtz Centre for Infection Research, Inhoffenstr. 7, 38124 Braunschweig, Germany. [6]Institute for Toxicology, Core Facility Proteomics, Hannover Medical School, Carl-Neuberg-Str. 1, 30625 Hannover, Germany. [7]Institute for Biochemistry, Biotechnology and Bioinformatics, Technische Universität Braunschweig, Spielmannstr. 7, 38106 Braunschweig, Germany. [8]These authors contributed equally: Sebastian Roman Borgert, Steffi Henke. ✉e-mail: wulf.blankenfeldt@helmholtz-hzi.de

Examples include toxic proteins and small organic molecules such as the phenazine derivative pyocyanin, a redox-active compound that causes oxidative stress and also acts as a respiratory pigment in anoxic environments[5]. Further, *P. aeruginosa* forms biofilms where it is protected from the immune system[4] and difficult to target by pharmaceuticals. The production of virulence factors and biofilm is controlled by cell-density-dependent signal transduction known as quorum sensing (QS). QS relies on small "autoinducer" signaling molecules that are secreted into the environment, from where they can be taken up by neighboring cells. Once a threshold concentration is reached, autoinducers bind and activate their cognate transcription factors. As a result, a defined regulon that often includes the respective autoinducer synthase itself is transcribed, creating a feedback loop that triggers virulence factor or biofilm formation in a switch-like manner.

The QS network of *P. aeruginosa* consists of at least three autoinducer circuits and also receives input from other agonistic and antagonistic systems[6]. In common with many other gram-negative bacteria, two QS circuits use *N*-acyl-L-homoserine lactones (HSLs) as autoinducers, namely *N*-(3-oxododecanoyl)-L-HSL (3-oxo-C12-HSL) and *N*-butanoyl-L-HSL (C4-HSL), which are produced by autoinducer synthases LasI and RhlI and activate the LuxR-type transcription factors LasR and RhlR, respectively. Unique to *P. aeruginosa* (and some strains of *Burkholderia*[7]) is the "*Pseudomonas* Quinolone System" (*pqs*), which employs 2-alkyl-4-quinolones (AQs) as autoinducers of a LysR-type transcription factor termed PqsR or MvfR[8,9].

The three QS systems are often regarded as hierarchical with LasI/LasR being at the top of the cascade to induce the expression of *pqs* and *rhl* genes (Fig. 1)[6]. However, it has been demonstrated that RhlR can act antagonistically to *pqs*[10] and that it can also induce genes controlled by LasR when the latter is missing[11], explaining why environmental or clinical isolates of late and chronic infections are often defective in LasI/LasR signaling[12,13]. While LasR and RhlR are known to regulate dozens of genes directly[14–16], PqsR seems to control the *pqs*-operon only[17]. The *pqs*-operon comprises genes *pqsA* to *pqsE*, which encode enzymes that biosynthesize 2-heptyl-4-quinolone (HHQ) and other AQs. PqsH, an oxidase outside of the *pqs*-operon, converts HHQ to 2-heptyl-3-hydroxy-4-quinolone, the "*Pseudomonas* Quinolone Signal" (PQS)[8]. HHQ and PQS both activate PqsR[18], but PQS has additional properties that lead to the induction of QS-independent processes[17].

Intriguingly, *pqsE* encodes a metal-dependent β-lactamase[19] whose role as a thioesterase in HHQ/PQS biosynthesis can be replaced by other enzymes encoded in the *P. aeruginosa* genome[20]. Despite this dispensability, however, deletion of *pqsE* leads to the loss of many QS-related traits and renders *P. aeruginosa* avirulent in infection models of *Caenorhabditis elegans* and mice[21,22]. PqsE was hence named "PQS response protein"[8], but it was shown later that its regulatory effect is due to an interplay with RhlR[23] that affects the majority of but not all RhlR-up- or down-regulated genes (Fig. 1)[16,22]. The molecular basis of this remains enigmatic, partially because of an intrinsic insolubility of RhlR that indicates impaired folding and hampers biochemical as well as structural studies. RhlR is not untypical in this regard, as related proteins often require HSLs for folding[24,25]. However, the autoinducer of RhlR, C4-HSL, alone is not sufficient to stabilize RhlR for isolation in significant amounts. Improvements towards generating soluble RhlR have recently been made through synthetic ligands such as 3-[4-(*m*-bromophenoxy)butyrylamino]-4,5-dihydro-3H-thiophen-2-one (*meta*-bromo-thiolactone, mBTL)[26] or by isolating stabilized RhlR variants in mutant screens[27]. Several studies indicate that the thioesterase activity of PqsE is not required in RhlR-mediated QS[28–32], and some mutations that impair its regulatory function are far from the active center[28–31]. Recent work suggested that PqsE may possess a second enzymatic activity that generates an unidentified compound to co-activate RhlR[22]. Lately, pull-down demonstrated that mBTL-stabilized RhlR interacts with PqsE and that this interaction is required for regulation[29,30,33].

In this work, we have identified this interaction in independent experiments and used it to isolate and characterize a PqsE/RhlR

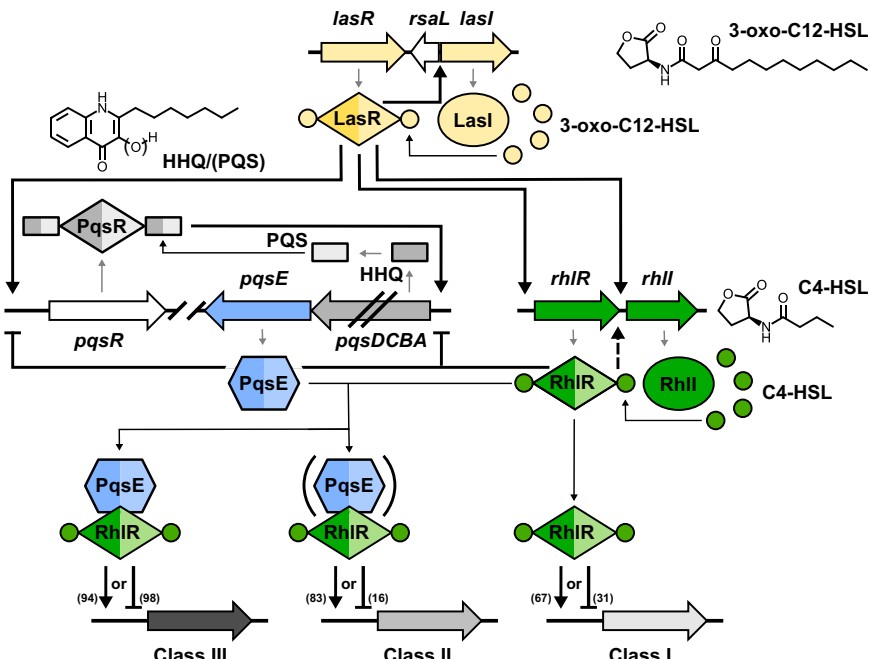

**Fig. 1 | The quorum sensing network of *P. aeruginosa*.** *P. aeruginosa* employs three QS circuits in a hierarchical, intertwined manner. The *las* system utilizes the homoserine lactone 3-oxo-C12-HSL as autoinducer for LasR, whereas the *rhl* and *pqs* systems use C4-HSL and 2-heptyl-4-quinolone (HHQ) or 2-heptyl-3-hydroxy-4-quinolone (PQS) to activate RhlR and PqsR (also termed MvfR), respectively. RhlR requires C4-HSL for up- or downregulation of target genes. The major part of the RhlR regulon is further influenced by association with PqsE, which is either required or at least enhances regulation by RhlR, leading to the indicated three sub-regulons class I-III. The number of up- or downregulated genes according to reference 16 is shown in parentheses[16]. LasR, RhlR, PqsR and PqsE are homodimeric proteins. Thick black arrows indicate upregulation (dashed lines: surrogate regulation), capped lines indicate antagonistic activity. Thin black arrows denote association; thin gray arrows represent biosynthetic processes.

complex with structural methods. Our data suggest that the importance of PqsE lies in providing additional stabilization to RhlR, hinting at chaperone-like activity that is specific to the QS network of *P. aeruginosa*.

## Results

### PqsE synergistically enhances the solubility of C4-HSL-stabilized RhlR

Because of the intrinsic insolubility of RhlR, the importance of HSL-autoinducers for the folding of many related transcription factors and the requirement for PqsE in the majority of RhlR-controlled processes, we reasoned that C4-HSL and PqsE may act together to increase the level of soluble active RhlR in bacteria producing the transcription factor. To corroborate this, we analyzed the soluble fraction of lysate from *E. coli* overexpressing RhlR at increasing concentrations of exogenously added C4-HSL in the absence or presence of PqsE

overexpression. Western blot analysis demonstrates that only minute amounts of soluble RhlR are detected without C4-HSL and that PqsE enhances RhlR solubility up to at least 50 μM C4-HSL, a concentration corresponding to the higher C4-HSL levels reported for *P. aeruginosa*[34–36]. The amount of RhlR in whole cell extracts, on the other hand, remained unchanged, showing that C4-HSL and PqsE indeed enhance the solubility but not the total amount of RhlR (Fig. 2a, Supplementary Fig. 1a). These observations may explain why C4-HSL was found as absolutely required in a recent study analyzing the RhlR-transcriptome of *P. aeruginosa*[16] and reflects the findings of previous work observing increased levels of RhlR in *P. aeruginosa* when PqsE was overexpressed[37,38]. For further consolidation, we used proteomic techniques to quantify the amount of soluble RhlR in *P. aeruginosa* and compared it to mutant strains unable to produce C4-HSL-autoinducer synthase RhlI and/or PqsE, which again pointed at a chaperone-like activity of PqsE that is synergistic to C4-HSL (Supplementary Fig. 1b).

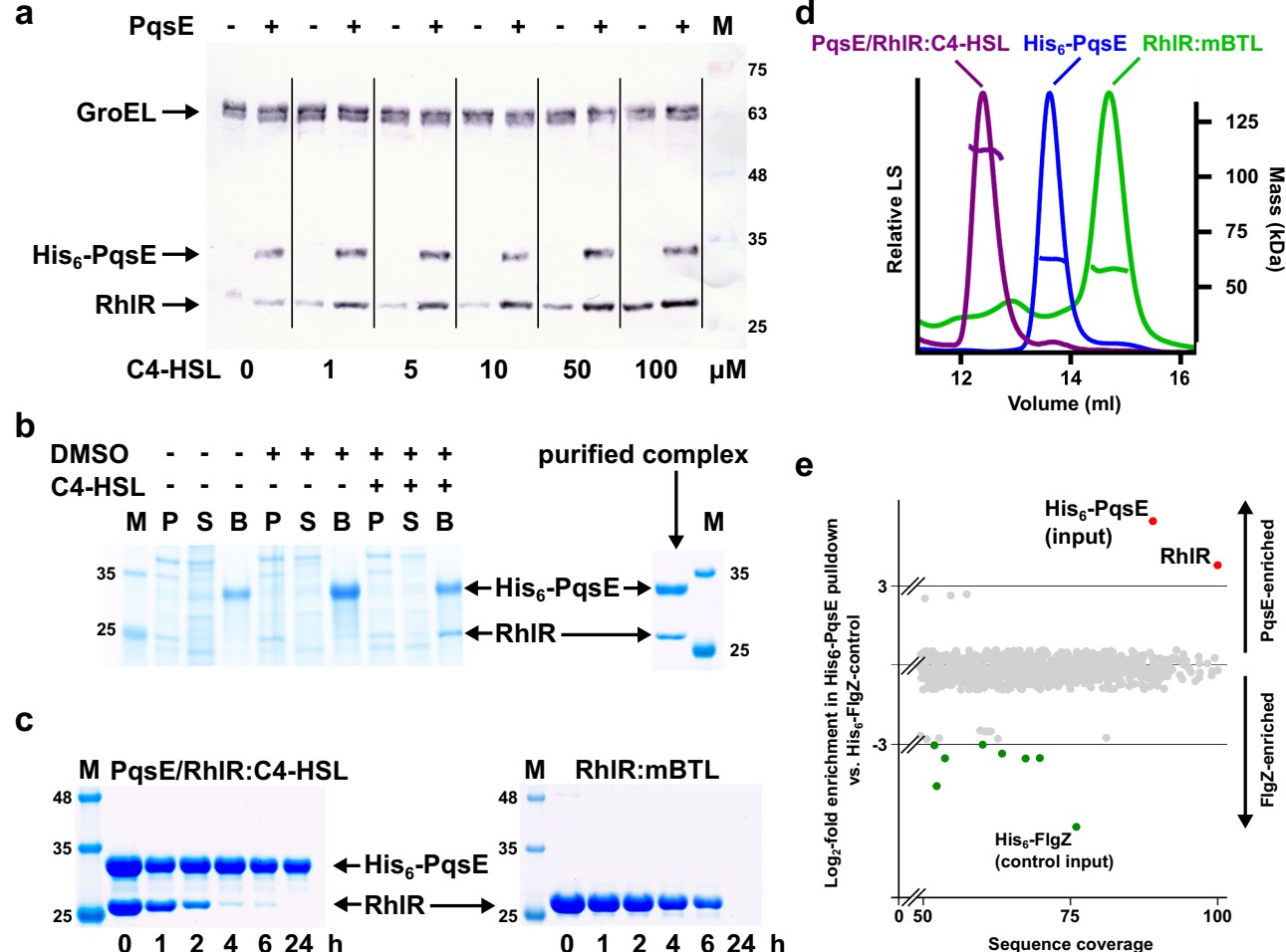

**Fig. 2 | Interaction of RhlR and PqsE. a** Western blot demonstrating the synergistic solubility-enhancing effect of C4-HSL and PqsE on RhlR. RhlR and His$_6$-PqsE were expressed from plasmid vectors in *E. coli* BL21-CodonPlus(DE3)-RIL in the presence of the indicated concentrations of C4-HSL. Soluble extracts were analyzed using GroEL as a control. The marker (M) indicates the molecular weight in kDa. A typical result from at least three independent experiments is shown. The corresponding whole-cell extracts are shown in Supplementary Fig. 1a. **b** SDS-PAGE of a co-expression analysis of His$_6$-tagged PqsE and RhlR in *E. coli* BL21-Codon-Plus(DE3)-RIL. Induced cells were harvested, lysed and separated into an insoluble (P) and a soluble fraction (S), which was used to incubate Ni-NTA beads, washed and then boiled in SDS-loading buffer (B). The marker (M) indicates the molecular weight in kDa. A typical result from at least three independent experiments is shown. **c** Time-dependent stability analysis of PqsE/RhlR:C4-HSL and RhlR:mBTL.

Purified proteins were incubated at 37 °C and the soluble fraction was analyzed by SDS-PAGE after the indicated period. The marker (M) indicates the molecular weight in kDa. A typical result from at least three independent experiments is shown. **d** SEC-MALS analysis of RhlR purified in the presence of mBTL, of PqsE and of the PqsE/RhlR:C4-HSL complex. The molecular mass of PqsE suggests that the protein is dimeric. PqsE and RhlR interact in a 2:2 complex. **e** Proteomic analysis of a pulldown with His$_6$-tagged PqsE in *P. aeruginosa* PA14 vs. a His$_6$-FlgZ control. Buffers were supplemented with 50 μM C4-HSL. Proteins with more than 50% sequence coverage in the analysis are shown. Log$_2$-fold enrichments larger than 3 (Ni-NTA bead eluate vs. whole-cell extract) are colored in red and green, respectively. The analysis confirms the interaction of PqsE and RhlR in *P. aeruginosa*. Source data are provided as a Source Data file.

It has previously been demonstrated that the solubility of RhlR can also be enhanced by mBTL[26], and we used this to isolate recombinant RhlR in the absence of PqsE. The synthetic HSL analogue mBTL has been developed as a RhlR antagonist, however, using pyocyanin production as a proxy for RhlR/PqsE-mediated quorum sensing, we observed that mBTL induced similar levels of pyocyanin as exogenously supplied C4-HSL in a Δ*rhlI* mutant of *P. aeruginosa* PA14 (Supplementary Fig. 1c), suggesting again that stabilization is a key factor of RhlR-mediated signaling.

## PqsE and RhlR form a 2:2 complex

Since several studies have shown that the enzymatic activity of PqsE is not required for regulating gene expression *via* RhlR and since the experiments above suggest that the importance of PqsE lies in solubility enhancement of RhlR, we hypothesized that the PqsE/RhlR interplay is rooted in a direct protein/protein interaction that aids RhlR to attain and maintain a folded and hence functional state. We, therefore, investigated whether both proteins engage in a complex when affinity-tagged variants are co-expressed in *E. coli*. This was indeed the case but required the presence of the cognate autoinducer C4-HSL or of the synthetic RhlR ligand mBTL (Fig. 2b). Isolation of the complex was possible, but only when the affinity tag was fused to the N-terminus of PqsE. Tagging of RhlR abrogated complex formation.

We then purified the PqsE/RhlR complex chromatographically (Fig. 2b) but initially failed to identify suitable crystallization conditions: whenever crystals were obtained, they contained only PqsE. This is in line with the observation that the complex was not stable, loosing RhlR by precipitation over time (Fig. 2c). We also did not succeed in stabilizing the complex by buffer optimization. Precipitation at a similar rate also affected the RhlR:mBTL complex (Fig. 2c). The precipitation tendency of RhlR may constitute a means to de-activate RhlR-mediated signaling *in cellulo* and it will be interesting to identify its underlying mechanistic basis in future studies. In addition to general fold instability, oxidative degradation owing to the three free cysteines within the RhlR monomer may be considered towards this.

Analysis of freshly purified complex by size-exclusion-chromatography multi-angle light scattering (SEC-MALS) revealed a 2:2 PqsE/RhlR stoichiometry (Fig. 2d). Interestingly, while the presence of two chains of RhlR was expected since similar transcription factors are homodimeric, PqsE itself was found to be dimeric as well (Fig. 2d). We, therefore, re-analyzed crystal structures of PqsE in the Protein Data Bank[39] and found that PqsE forms similar dimers in all of these entries (Supplementary Fig. 2a). After introducing a mutation in the dimer interface (E187R), we could obtain and crystallize a monomeric form of PqsE (Supplementary Fig. 2b). The mutation led to strongly reduced production of the PqsE/RhlR-controlled phenazine virulence factor pyocyanin (Supplementary Fig. 2c) and also weakened the interaction with RhlR significantly (Supplementary Fig. 2d), suggesting that dimerization of PqsE is a prerequisite for tight interaction with RhlR. Impaired dimer formation may also explain the pyocyanin-reducing effect of mutations within α-helices at the C-terminus of PqsE that have recently been observed by others[30,31]. These α-helices belong to the dimerization motif of PqsE and are unique within the metallo-β-lactamase family[19]. The finding that both RhlR and PqsE are homodimers is expected to govern the relative orientation of both proteins in the complex because of the likely alignment of their symmetry axes.

To confirm that the formation of a PqsE/RhlR complex is not an artefact arising from overexpression in *E. coli*, we performed pull-down experiments in *P. aeruginosa* PA14. Here, we introduced a plasmid-encoded His$_6$-tagged variant of PqsE into a Δ*pqsE*-deleted strain, isolated protein on IMAC beads and identified co-purified proteins by LC-MS. This revealed significant enrichment (approx. 14-fold) with high sequence coverage for RhlR versus a His$_6$-FlgZ control (Fig. 2e).

Finally, we employed microscalar thermophoresis (MST) to quantify the interaction between PqsE and RhlR purified in the presence of mBTL. Here, we measured a dissociation constant of $K_D = 146 \pm 2$ nM (Supplementary Fig. 2d), hinting at considerable affinity between both proteins. Similar experiments with PqsE-E187R were more difficult to perform due to rapid protein precipitation and yielded an inverse titration curve corresponding to a $K_D$ of $13.8 \pm 3.9$ μM (Supplementary Fig. 2d).

## Structure of the PqsE/RhlR complex

Because initial attempts to identify crystallization conditions for the PqsE/RhlR complex failed, we first resorted to peptide SPOT arrays[40] to gain more insight into the interaction between both proteins. Here, 21-mers covering *P. aeruginosa* PA14/PAO1 RhlR in steps of three amino acids (Supplementary Table 9) were incubated with His$_6$-tagged PqsE and anti-His$_6$-tag antibody was used to detect immobilized PqsE. While these array data were one of our first indications for an interaction of both proteins, these experiments suffered from high background in the negative control, hampering straightforward interpretation (Supplementary Fig. 3a).

Next, we attempted to stabilize the PqsE/RhlR complex by generating a fusion protein in which the N-terminus of RhlR was fused to the C-terminus of PqsE *via* a flexible XTEN-type linker[41] of 30 residues length (Supplementary Fig. 3b). This protein restored pyocyanin production in a Δ*pqsE*/Δ*rhlR*/Δ*rhlI* triple mutant *P. aeruginosa* PA14 in a C4-HSL- or mBTL-dependent manner, showing that it acts as a surrogate of the separated proteins (Supplementary Fig. 3c). Recombinant production of the fusion protein was straightforward but required C4-HSL or mBTL, reflecting the stability-enhancing effect of these ligands. The fusion protein in the presence of mBTL crystallized in space group I422 with one homodimer in the asymmetric unit, but the linker between PqsE and RhlR was flexible and did not yield visible electron density, whereas mBTL was found in the ligand binding domain (LBD) of RhlR (Supplementary Fig. 3b).

We then used the optimized precipitant for the fusion protein to crystallize freshly purified PqsE/RhlR complex in the presence of C4-HSL or mBTL and obtained crystals with similar unit cell dimensions that diffracted anisotropically up to 3.06 Å in the a* and b*, but only to approx. 4 Å resolution in the c* direction (cutoff criterion I/σ(I) ≥ 1.2 in the highest resolution shell; Supplementary Table 7). The relative orientation of PqsE and RhlR in these structures was identical to that observed in XTEN30-fusion protein (Fig. 3a; Supplementary Fig. 3b; Supplementary Fig. 4a), and since no significant differences between the PqsE/RhlR:C4-HSL and the PqsE/RhlR:mBTL complex were observed (RMSD = 0.21 Å for 1080 of 1085 residues), we discuss the C4-HSL-bound complex here.

The PqsE segment adopted the same dimeric configuration as in the free protein but with a slightly twisted arrangement (Fig. 3a). The RhlR sequence displayed the expected homodimeric structure of a LuxR-type transcription factor and C4-HSL as well as mBTL were found in the anticipated ligand binding site (Fig. 3a, Supplementary Fig. 4a/b). Interactions between PqsE and RhlR do not involve the autoinducer binding site or the DNA-binding domain (DBD) of RhlR, suggesting that complex formation does not directly interfere with the ability of RhlR to sense C4-HSL and bind to its cognate promoter regions. The homoserine lactone moiety of C4-HSL establishes a hydrogen bond with the side chain of W68, whereas the amide group engages into hydrogen bonds with Y64, D81 and S135 (Fig. 3a). These interactions are also found in HSL complexes of other LuxR-type transcription factors such as e.g. *P. aeruginosa* LasR in complex with 3-oxo-C12-HSL (PDB entry 2UV0)[42]. The acyl chain of C4-HSL points into a hydrophobic pocket of the ligand binding site in RhlR, again with high similarity to related structures. This pocket extends beyond the butyryl group of C4-HSL and is not fully closed towards the solvent, explaining why ligands such as mBTL can be accommodated without

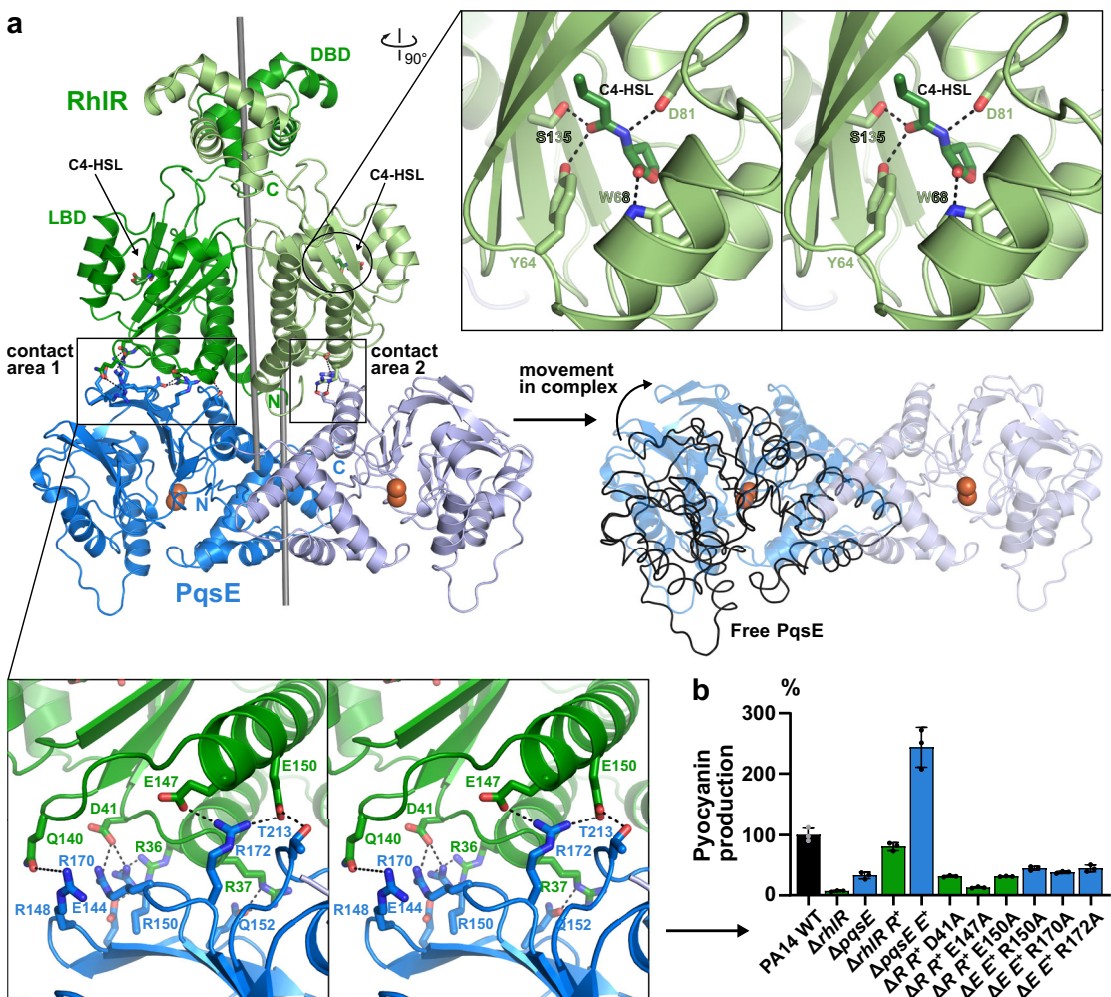

**Fig. 3 | Structure of the PqsE/RhlR complex. a** Crystal structure of the PqsE/RhlR:C4-HSL complex. DBD and LBD denote the DNA- and ligand-binding domains of RhlR, the two-fold symmetry axes of RhlR and PqsE are shown as sticks (calculated with draw_rotation_axis.py in PyMOL[73]). Red spheres represent two iron atoms bound to the active center of PqsE. The magnified area at the top right shows a cross-eyed stereo plot of polar interactions between C4-HSL and RhlR, the magnified area below demonstrates polar interactions between RhlR and PqsE in the larger contact area 1. Complex formation with RhlR distorts the PqsE dimer with respect to the free enzyme (black lines, middle-right panel), leading to movements of more than 10 Å at extreme positions. **b** Mutation of selected residues in contact area 1 impairs the production of pyocyanin in pUCP24-supplemented (indicated by R$^+$ or E$^+$, respectively) Δ*rhlR* and Δ*pqsE* mutants of *P. aeruginosa* PA14. Mean values with error bars indicating the standard deviation of three independent measurements are shown. Source data are provided as a Source Data file.

obvious changes to the structure (Supplementary Fig. 4a) and suggesting that RhlR could sense other HSLs as well. Additional interactions of mBTL are described below.

Notably, only the N-terminus of PqsE is accessible in the complex, explaining why tagging at other termini abrogates complex formation. The complex is only pseudo-symmetric, i.e. the twofold axes of the PqsE and RhlR proteins are slightly shifted and skewed relative to each other (Fig. 3a). This leads to the formation of two different contacts between both proteins (Fig. 3a), one comprising approx. 850 Å$^2$ (contact area 1) and a smaller one involving approx. 310 Å$^2$ (contact area 2) partially overlapping with contact area 1. Both contact areas involve polar residues (Fig. 3a). To validate their relevance, we introduced alanine mutations and investigated their impact on complex formation in *E. coli* coexpression experiments as well as on pyocyanin production in *P. aeruginosa*. This revealed the importance of several (e.g. RhlR: D41, E147, E150; PqsE: R150, R170, R172) but not all residues (e.g. RhlR: Q140/Q141; PqsE: R148) within the larger contact area 1, whereas mutation of residues exclusive to contact area 2 had no effect (Fig. 3b, Supplementary Fig. 4c). Interestingly, the electron density of the PqsE monomer involved in contact area 1 was of much lower quality than for the other chains of the complex. It is presently unclear

if this is a crystallographic artefact resulting from anisotropic diffraction or a consequence of distortion within the PqsE dimer upon interaction with RhlR (Fig. 3a).

## PqsE- and C4-HSL-independent RhlR signaling

The observation that both PqsE and C4-HSL together increase the amount of soluble RhlR suggest that they control RhlR by enhancing its stability. To corroborate this further, we sought to replace wildtype RhlR with an in-silico optimized variant that triggers pyocyanin production independently of PqsE and C4-HSL. Here, we used the PROSS (Protein Repair One-Stop Shop) webserver[43] to design RhlR derivatives with higher stability. Indeed, the PROSS-designed RhlR-P75, in which 75 (out of 241) residues were mutated with respect to the original sequence, could easily be produced in *E. coli*, even in the absence of C4-HSL or mBTL. The variant displayed a melting temperature of 84.3 ± 1.3 °C in the absence of ligands, which increased to over 90 °C in buffer containing 1 mM mBTL, nearly 40 K higher than the WT protein under similar conditions (Fig. 4a). It crystallized readily in the presence of C4-HSL or mBTL, but not without these ligands. A complex with mBTL diffracted to 2.15 Å resolution and yielded well-resolved electron density for the ligand (Fig. 4b, Supplementary Fig. 5a).

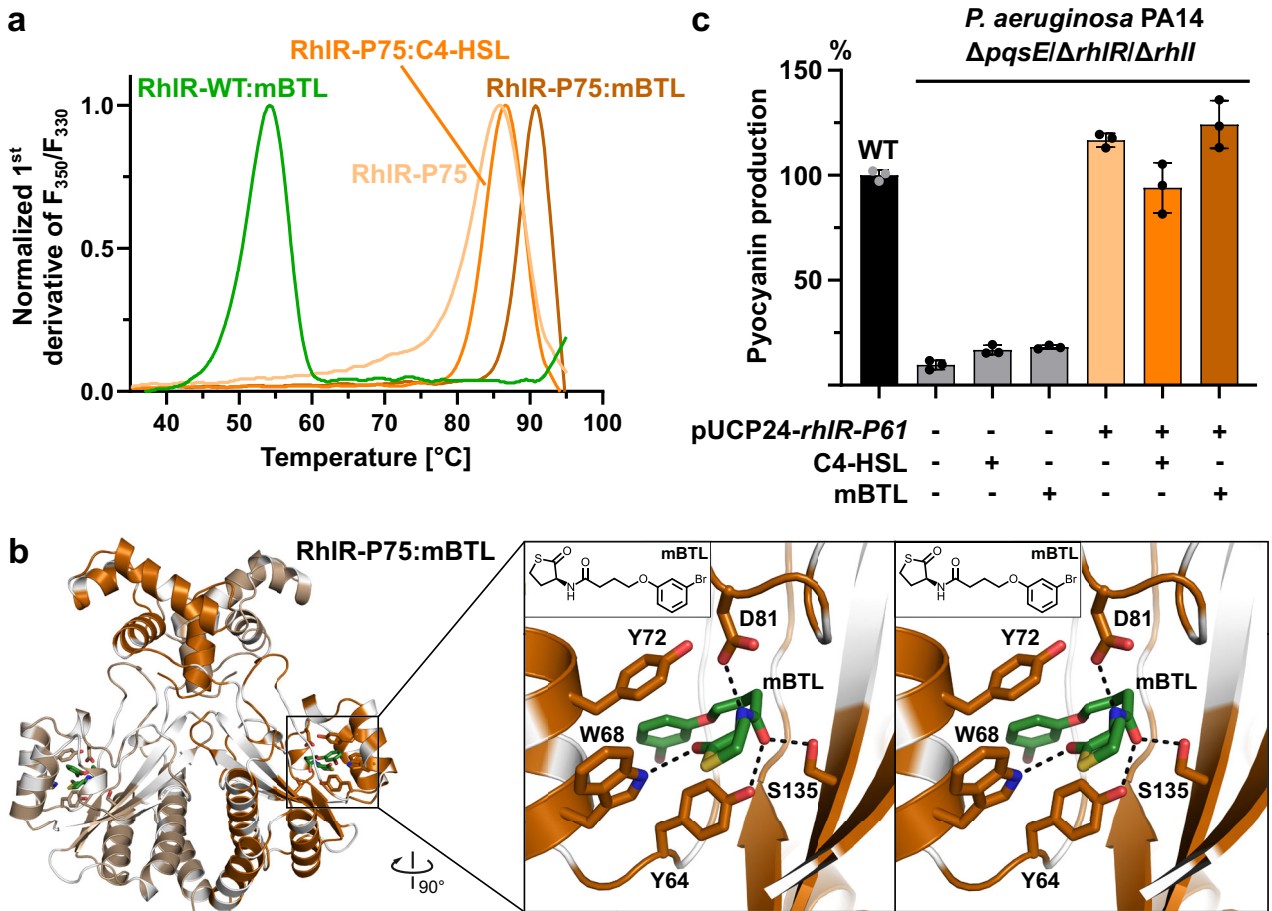

**Fig. 4 | A stability-optimized variant of RhlR overcomes PqsE- and C4-HSL-dependent RhlR-signaling. a** Protein melting point analysis of the stability-optimized RhlR-P75 vs. wildtype RhlR in the presence or absence of C4-HSL or mBTL. **b** Crystal structure of RhlR-P75 in complex with mBTL. Positions that have been altered with respect to the wildtype protein are shown in white. The magnified insert shows a cross-eyed stereo plot of mBTL in the ligand binding site.

**c** Stability-optimized RhlR-P61 (RhlR-P75 with WT-DBD) stimulates pyocyanin production in a *P. aeruginosa* PA14 Δ*rhlR*/*rhlI*/*pqsE* triple mutant without requiring C4-HSL or mBTL. Mean values with error bars indicating the standard deviation of three independent measurements are shown. Source data are provided as a Source Data file.

Identical to the less well-resolved PqsE/RhlR:C4-HSL and PqsE/RhlR:mBTL complexes described above, the thiolactone is embedded in a mainly hydrophobic pocket and establishes hydrogen bonds with Y64, W68, D81 and S135. The butanol-*m*-bromophenol group of mBTL resides in the hydrophobic acyl-binding tunnel leading towards the lactone binding site, and the *m*-bromophenol establishes π-stacking interactions with Y64 and Y72 (Fig. 4b), two residues also contained in wildtype RhlR. These interactions may lead to tighter binding and explain why mBTL enhances the stability of RhlR more than the cognate autoinducer C4-HSL[26]. Of note, the conformation of mBTL differs from that observed in a complex with LasR[44] (Supplementary Fig. 5b). A complex with C4-HSL was less well resolved (3.5 Å resolution), but the shape of the electron density confirms similar interactions (Supplementary Fig. 5c).

Co-expression of RhlR-P75 and His₆-tagged PqsE revealed that the interaction between both proteins has been lost by the introduced mutations. Loss of interaction was also found for variant RhlR-P61 (Supplementary Fig. 5d), in which we back-mutated residues in the DNA-binding domain to the wildtype sequence to assess the RhlR-regulatory activity of the stabilized protein. We then used a RhlR-P61-encoding plasmid vector to transform a Δ*pqsE*/Δ*rhlR*/Δ*rhlI* triple mutant of *P. aeruginosa* PA14. Here, the production of pyocyanin was restored to wildtype levels and not significantly altered by C4-HSL or mBTL, demonstrating that the stabilized RhlR indeed acts independently of PqsE and of these ligands (Fig. 4c). Crystallization of RhlR-P61

with mBTL showed the ligand in a similar position and conformation as in the other crystal structures determined here (Supplementary Fig. 5e).

## Discussion

Quorum sensing (QS) is a seemingly simple mechanism that allows bacteria to control gene expression in a cell-density-dependent manner and hence act in a coordinated fashion. Complexity results from the intertwining of several QS circuits, interfacing with other regulatory inputs (also including QS signals from other bacteria) and from the addition of uncommon regulatory mechanisms, together resulting in species-specific regulatory networks that can react to a multitude of situations in a switchboard-like manner.

*P. aeruginosa* possesses one of the best-studied QS cascades[6], which also reflects the clinical relevance of *P. aeruginosa* as an opportunistic pathogen. Since disarming bacteria of their virulence factors instead of attacking them with bactericidal agents is expected to avoid resistance development[45] and since virulence is often QS-controlled, QS is an attractive target for novel antimicrobial approaches. Accordingly, the unique *pqs* system of *P. aeruginosa* has received significant attention for "pathoblocker" development recently, mostly by addressing the PQS-specific transcription factor PqsR/MvfR[46–52]. However, because only the *pqs*-operon is directly regulated by PqsR[17], we hypothesize that PqsR-targeting pathoblockers act rather indirectly by reducing the production of PqsE and thereby impacting on RhlR,

the major regulator of QS-mediated virulence. It may hence be more effective to address the PqsE-mediated RhlR-regulation downstream of PQS itself.

PqsE is a thioesterase that triggers transcription of the RhlR regulon through a hitherto cryptic mechanism, which we identify here as involving a significant chaperone-like component that enhances the stability of RhlR *via* protein complex formation. While this moonlighting activity of the enzyme PqsE is surprising, the activation of other related LuxR-like transcription factors through enhanced folding by genuine chaperones was described previously, e.g. for the production of *Vibrio fischeri* LuxR in the heterologous host *E. coli*[53]. Yet, the effect of PqsE on RhlR is, to our knowledge, the first example of a solubility-enhancing interaction on such a transcription factor within the native host.

Coexpression experiments in *E. coli* indicated that the cognate autoinducer C4-HSL is absolutely required for producing significant amounts of soluble RhlR and that high concentrations may even overdrive the importance of PqsE (Fig. 2a). In this regard, it is interesting to note that not all RhlR-controlled processes depend on PqsE[16,22]. A well-known example is the biosynthesis of rhamnolipids, which is not significantly downregulated in ΔpqsE mutants of *P. aeruginosa*[23,38]. We speculate that the respective DNA promoter sequences possess higher affinity for and can hence sense lower levels of RhlR than those that control e.g. pyocyanin production, a process that strictly depends on both C4-HSL and PqsE. The observation that relevant levels of C4-HSL lead to detectable amounts of soluble RhlR (Fig. 2a) seem to support this hypothesis, but further experiments are required for corroboration.

While we cannot exclude that C4-HSL and PqsE have allosteric effects that contribute to fine-tuning the DNA-binding properties of RhlR in a similar manner as to what has been reported for autoinducer binding of related transcription factors such as *V. fischeri* LuxR[54] or *P. aeruginosa* QscR[55], our findings suggest that stability enhancement by complex formation with C4-HSL and PqsE plays a major role in RhlR-mediated regulation. This can also be concluded from the observation that the addition of mBTL, a compound developed to antagonize but also found to stabilize RhlR[26], led to similar levels of pyocyanin as C4-HSL in a ΔrhlI mutant of *P. aeruginosa* PA14 in our experiments (Supplementary Fig. 1c). The complex structures obtained here may provide a starting point to develop these HSL analogues further, however it may ultimately be difficult to find clear-cut antagonists of RhlR, since such compounds are expected to stabilize and hence activate RhlR at least to some extent. Instead, it may be more promising to leverage our insight into the interaction with PqsE for the development of pathoblockers that target this *P. aeruginosa*-specific QS mechanism.

## Methods

### Reagents
All chemicals, primers, reagents for molecular biology, synthetic genes, chromatography materials, plasticware and other consumables were purchased from commercial sources. *Pseudomonas aeruginosa* PA14 was obtained from a previously published collection of transposon mutants[56].

### Construction of *P. aeruginosa* mutants
ΔpqsE, ΔrhlR and ΔpqsE/ΔrhlR deletion mutants of *P. aeruginosa* PA14 were obtained from a transposon[56] library or generated by recombination as described previously[57].

Deletions of *rhlI* were generated with a self-developed CRISPR/Cas9-selection, Ssr-mediated recombination method.[58] Briefly, plasmid pS448•CsR, developed for genome engineering in *P. putida*[59], was used for constitutive expression of a target-specific small guide RNA (sgRNA) and the 3-methylbenzoate-inducible production of the Cas9 endonuclease. A second vector, based on pSEVA624 and encoding

IPTG-inducible Ssr recombinase from *P. putida* DOT-T1E[59], allowed recombination with synthetic linear DNA. Successful deletion was confirmed by colony PCR and mutants were finally cured from plasmids by three consecutive passages in antibiotic-free LB media. Plasmid loss was confirmed by selection with the respective antibiotics.

Deletion mutants were complemented by transformation with plasmid pUCP20 or pUCP24 into which the respective gene had been cloned by standard techniques.

Primers, plasmids and strains used for the construction of the *P. aeruginosa* PA14 derivatives used in this study are listed Supplementary Tables 1–3.

### Synthesis of C4-HSL and mBTL
C4-HSL[60] and mBTL[26] were synthesized and purified at the multigram-scale by following published procedures. Their identity and purity were confirmed by mass spectrometry and NMR spectroscopy as listed below.

Briefly, C4-HSL was obtained by dropwise addition of butanoyl chloride (1.21 mL, 10.6 mmol, 1.00 eq.) at room temperature to a magnetically stirred solution of *L*-(-)-α-amino-γ-butyro lactone hydrobromide (2.50 g, 13.7 mmol, 1.30 eq.) and sodium carbonate (2.90 g, 27.5 mmol, 2.60 eq.) in a biphasic mixture of methylene chloride and water (110 mL, 1:1). Stirring was continued at the same temperature for 2.5 h. The reaction was terminated by dilution with methylene chloride (50 mL) before the aqueous and organic layers were separated. The aqueous layer was extracted with methylene chloride (3 × 20 mL) and the combined organic layers were washed with aqueous sodium bicarbonate (2 × 20 mL) and dried over anhydrous sodium sulfate. Evaporation of all volatiles and drying in vacuo furnishes pure C4-HSL (1.85 g, 10.81 mmol, 79% yield).

**$^1$H-NMR** (400 MHz, CDCl$_3$) δ [ppm]: 5.96 (1H, s, broad), 4.59-4.52 (1H, m), 4.48 (1H, td, $J$ = 1.0, 9.6 Hz), 4.32-4.24 (1H, m), 2.92-2.84 (dddd, 1H, $J$ = 1.0, 5.8, 8.4, 12.6 Hz), 2.24 (2H, dt, $J$ = 1.4, 7.3 Hz), 2.19-2.06 (1H, m), 1.69 (2H, sext, $J$ = 7.3 Hz), 0.96 (3H, t, $J$ = 7.5 Hz). **$^{13}$C-NMR** (100 MHz, CDCl$_3$) δ [ppm]: 175.7, 173.7, 66.2, 49.3, 38.1, 30.7, 19.0, 13.8. **HRMS** (ESI +) m/z calculated for $C_8H_{13}NO_3Na^+$ [M + Na]$^+$ 194.0793, found 194.0793.

mBTL was synthesized *via* ethyl 4-(3-bromophenoxy)butanoate, which was then hydrolyzed to 4-(3-bromophenoxy)butanoic acid before amidation to mBTL. For this, to a solution of 3-bromophenol (6.5 g, 37.6 mmol) in DMF (34.2 ml), cesium carbonate (36.7 g, 113 mmol) and ethyl 4-bromobutanoate (5.65 ml, 37.7 mmol) were added sequentially at room temperature. The heterogeneous reaction mixture was stirred at room temperature overnight and then partitioned between ether (300 mL) and water (200 mL). The ether layer was collected, washed with brine (200 mL), dried over sodium sulfate and concentrated to yield ethyl 4-(3-bromophenoxy)butanoate (10.32 g, 35.9 mmol, 96% yield) as an oil. This was added to a solution of sodium hydroxide (7.17 g, 179 mmol) in THF (90 ml) with water (29.9 ml) and heated at 65 °C overnight. The reaction mixture was acidified to pH 2 with 1 M HCl. The aqueous layer was extracted with 3 × 100 ml EtOAc. The organic layer was washed with brine and dried with Na$_2$SO$_4$ to give 4-(3-bromophenoxy)butanoic acid (8.8 g, 34.0 mmol, 95% yield) as a yellow syrup. This was added to a flame-dried flask together with *N*-(3-dimethylaminopropyl)-*N*'-ethylcarbodiimide hydrochloride (7.16 g, 37.4 mmol), 1-hydroxybenzotriazole (1.30 g, 8.49 mmol), triethylamine (4.3 mL, 34 mmol), homocysteine thiolactone hydrochloride (6.46 g, 34.0 mmol) and CH$_2$Cl$_2$ (340 mL). After the mixture was stirred at room temperature for 24 h, water was added and the aqueous layer was extracted with 3 × 250 mL EtOAc. The combined organic layer was washed sequentially with 1 M NaHSO$_4$ (250 mL), saturated aqueous NaHCO$_3$ (250 mL), and brine (250 mL). The solution was dried over Na$_2$SO$_4$, filtered, and concentrated. The crude product was purified by column chromatography (PE /ETOAc: 20–60%) to yield mBTL (4.5 g, 35.4% yield) as a white solid (32.3% overall yield).

**¹H NMR** (400 MHz, CDCl$_3$) δ [ppm]: 7.18–6.94 (3H, m), 6.79 (1H, ddd, $J$ = 8.2, 2.4, 1.1 Hz), 6.25 (1H, d, $J$ = 6.3 Hz), 4.54 (1 Hz, dt, $J$ = 13.1, 6.7 Hz), 3.96 (2H, td, $J$ = 6.1, 1.2 Hz), 3.33 (1H, m), 3.21 (1H, m), 2.91–2.76 (1H, m), 2.49–2.34 (2H, m), 2.19–2.05 (2H, m), 1.91 (1H, m). **¹³C NMR** (100 MHz, CDCl$_3$) δ [ppm]: 205.58, 172.82, 159.58, 130.61, 123.88, 122.80, 117.82, 113.48, 67.05, 59.39, 32.45, 31.79, 27.54, 24.89. **HRMS** (ESI + ) m/z calculated for $C_{14}H_{16}BrNO_3SNa^+$ [M + Na]$^+$ 379.9932, found 379.9937.

### Recombinant proteins

Overexpression was achieved from pET-based plasmid vectors, which were generated with PCR-amplified genes and standard molecular biology techniques (Supplementary Table 2). Plasmids for the expression of RhlR-P75 and RhlR-P61 were purchased from GenScript Biotech Corporation. Site-directed mutagenesis was performed with the QuikChange protocol.

Recombinant proteins were produced in heat-shock transformed *E. coli* BL21 CodonPlus (DE3), which were grown in ZYM-5052 or TB-5052 autoinduction media[61] for 24 h at 16 °C supplemented with 50 μM C4-HSL or mBTL. Cells were harvested by centrifugation and stored at −20 °C until needed. Thawed cell pellets were resuspended in lysis buffer (Supplementary Table 4) supplemented with protease inhibitors (cOmplete mini EDTA-free, Sigma Aldrich) and lysed by sonication. The lysate was clarified by centrifugation (36.000 × $g$ for 45 min and again at 100.000 × $g$ for 60 min) and then applied to a Ni$^{2+}$-charged HisTrap HP (Cytiva) or Strep-Tactin HC column (IBA) connected to an ÄKTApurifier or ÄKTA pure FPLC system (Cytiva). Bound proteins were eluted with elution buffer (Supplementary Table 4) and fractions containing pure protein as indicated by SDS-PAGE were pooled, incubated with TEV protease to remove the affinity tag and then dialyzed overnight at 4 °C against dialysis buffer (Supplementary Table 4). Uncleaved protein was separated by passing over the respective affinity column again and collecting the flowthrough, which was then concentrated by ultrafiltration before polishing by size-exclusion chromatography using a HiLoad 26/600 Superdex 200 column (Cytiva) equilibrated in SEC buffer (Supplementary Table 4). Fractions containing pure protein were again concentrated by ultrafiltration and the purified protein was either used immediately or stored at −80 °C until further usage. The PqsE-XTEN30-RhlR fusion protein and the PqsE/RhlR complex required an anion exchange step (Capto Q, Cytiva) to remove bound DNA before the SEC step.

### Assessment of pyocyanin production in *P. aeruginosa*

Pyocyanin was quantified using a previously published procedure[22]. Briefly, *P. aeruginosa* PA14 WT or mutant overnight cultures in LB medium at 37 °C were diluted 1:1000 into 2.5 ml of fresh medium. OD$_{600}$ was determined after 16 h of shaking at 37 °C before pelleting cells by centrifugation and measuring OD$_{695}$ of the supernatant in a microplate reader (Tecan Infinite M200) to report the ratio of OD$_{695}$/OD$_{600}$.

### Western blot analysis of RhlR solubility in *E. coli*

His$_6$-tagged PqsE (integrated into plasmid vector pET19m) and untagged RhlR (integrated into pET26b) were co-expressed in *E. coli* BL21 CodonPlus (DE3) in the presence of different concentrations of C4-HSL for 24 h at 16 °C using 50 ml ZYM-5052 autoinduction media[61]. Cells were harvested by centrifugation and stored at −20 °C until needed. Equal amounts of cells were resuspended in 2 ml lysis buffer and lysed by sonication. Whole cell extracts were prepared by incubating lysate with SDS loading buffer, soluble fractions were prepared by centrifugation of the lysate at 21.000 × $g$ for 45 min before mixing supernatant with SDS-loading buffer. Protein concentrations were determined by Bradford assay to dilute samples to equal levels. Samples were analyzed with western blots using α-RhlR (generated by Davids Biotechnologie GmbH using recombinant RhlR purified in the presence of mBTL, f.c. 1:33.333), α-PqsE (generated by BioGenes GmbH, f.c. 1:5.000) or α-groEL (Abcam ab90522, f.c. 1:10.000) as primary antibodies and alkaline phosphatase conjugated α-rabbit antibody (Promega S3731, f.c. 1:10.000) as secondary antibody (Supplementary Table 5).

### Proteomic analysis of RhlR solubility in *P. aeruginosa* PA14

*P. aeruginosa* PA14 wildtype and deletion mutants Δ*pqsE*, Δ*rhlR*, Δ*rhlI* and Δ*pqsE/rhlI* were freshly streaked on LB-agar. Colonies were grown in 3 ml TSB overnight for inoculation of 20 ml media shaken at 37 °C for 24 h. Cells were pelleted and washed three times in PBS before equal amounts of cells were harvested by centrifugation. Soluble fractions were prepared by suspending in 50 mM Bicine pH 8.0, 150 mM NaCl, 5% glycerol with protease inhibitors (cOmplete, EDTA-free) followed by sonication and centrifugation at 100.000 × $g$ for 30 min. Whole cell extracts were generated by lysis in SDS-containing buffer (2% SDS, 500 mM TEAB, 200 mM NaCl) with protease inhibitors (cOmplete, EDTA-free). Protein digestion and peptide purification for LC-MS/MS was performed using a slightly adapted SP3 protocol[62]. In short, proteins were subjected to reduction and alkylation using 5 mM TCEP and 10 mM MMTS, respectively. Proteins were allowed to bind to SP3 carboxylate beads overnight in the presence of 60% acetonitrile. Then, beads were washed twice with 70% ethanol and subsequently once with 100% acetonitrile. Proteins were digested overnight at 37 °C using trypsin at a final concentration of 1 μg protease to 50 μg total protein in the presence of 50 mM TEAB, 5 mM TCEP and 10 mM MMTS. Digested peptides were then allowed to bind to SP3 carboxylate beads overnight in the presence of 95% acetonitrile. After extensively washing the beads with 100% acetonitrile, peptides were eluted using 2% DMSO and milliQ H$_2$O. Samples were vacuum dried and resuspended in 0.1% folic acid before applying to C18 Evotips (EV-2001; Evosep) according to the manufacturer's protocol. Evotips were then loaded on an Evosep One HPLC (Evosep) connected to a TimsTOFPro mass spectrometer (Bruker) equipped with PaSER Version 2022c (Bruker) for real-time database searches. The Evosep One HPLC was operated with the standard 60 sample per day method (21 min gradient at a flow rate of 1.0 μl/min; buffer A: 0.1% formic acid, buffer B: 0.1% formic acid in acetonitrile). For the TimsTOF*Pro* mass spectrometer, the standard MSMS Bruker method "DDA PASEF method for short gradients with 0.5 s cycletime" was employed. MS settings in detail were: scan begin 100 m/z; scan end 1700 m/z; ion polarity: positive; scan mode: PASEF. Tims settings were: mode custom; number of PASEF ramps: 4; charge minimum: 0; charge maximum: 5; 1/$K_0$ start 0.75 V*s/cm$^2$; 1/$K_0$ end 1.4 V*s/cm$^2$; ramp time: 100.0 ms; MS average: 1. Settings for PaSER were fragmentation/activation method were CID/HCD; precursor/peptide mass tolerance: 20.0 ppm; precursor mass range: 400.0–6000.0 m/z; fragment mass tolerance: 50.0 ppm; enzyme: trypsin (K/R at C-terminus); maximum miscleavages: 1; static modification: 57.02146 on cysteine; differential modification: 15.99491 on methionine; database: *Pseudomonas aeruginosa* PA14 release 06 June 2021. For quantification the ID_STAT©COMPARE plug-in in PaSER Version 2022c was used. Data analysis, data presentation and statistics were performing using GraphPad Prism 9.3.1.

Experiments were performed in three independent biological replicates ($n$ = 3, Student's $t$-test).

### PqsE/RhlR interaction analysis by pulldown from *E. coli* lysate

His$_6$-tagged PqsE (integrated into plasmid vector pET19m) and untagged RhlR (integrated into pET26b) or their respective variants were co-expressed in *E. coli* BL21 CodonPlus (DE3) for 24 h at 16 °C using 50 ml ZYM-5052 autoinduction media[61] supplemented with 50 μM C4-HSL or mBTL. 10 ml cell culture were harvested by centrifugation and stored at −20 °C until needed. Cells were resuspended in 2 ml lysis buffer supplemented with 50 μM C4-HSL or mBTL and lysed by sonication. The soluble fraction was isolated by centrifugation at 17.000 × $g$ for 45 min

and incubated with 25 µl Ni-NTA beads (Qiagen) for 30 min. Beads were carefully washed with lysis buffer for 3 times prior to adding SDS loading buffer. Samples were analyzed by SDS.

## Stability analysis of PqsE/RhlR:C4-HSL and RhlR:mBTL

Recombinant proteins were brought to a concentration of 5 mg/ml in buffer containing 20 mM sodium phosphate pH 7.6, 150 mM NaCl and 50 µM C4-HSL or mBTL, respectively. 100 µl sample were incubated at 37 °C and 10 µl were removed after intervals of 0, 1, 2, 4, 6 and 24 h for immediate centrifugation at $21.000 \times g$ for 15 min. Supernatants were mixed with SDS sample buffer (ratio 1:5), incubated at 95 °C for 10 min and then analyzed by SDS-PAGE.

## Size-exclusion chromatography – multi-angle light scattering (SEC-MALS)

Experiments were performed on an Agilent 1260 Infinity II HPLC system equipped with a Superdex 200 Increase 10/300 or 5/150 column (Cytiva), a miniDAWN TREOS MALS detector, and an Optilab T-rEX refractometer (Wyatt Technology Corp.). The column was equilibrated in SEC buffer and 40–150 µg of protein were applied after 0.1 µm filtration. Data were processed with the Astra software package (Wyatt Technology Corp.).

## Identification of PqsE-interacting proteins in *P. aeruginosa*

Interaction partners of PqsE in *P. aeruginosa* were identified by pull-down combined with untargeted LC-MS proteomic analysis. Overnight cultures of PA14 tn*pqsE* pUCP20::His$_6$-*pqsE* and the control strain (PA14 Δ20700ΩGm pUCP20::His$_6$-flgZ) were used to inoculate a 100 ml LB culture to an OD$_{600}$ of 0.05, which was then shaken at 37 °C for 8 h (180 rpm). Cells were harvested by centrifugation, washed on ice with 2.5 ml pellet wash buffer (50 mM Tris, 300 mM NaCl, 20 mM imidazole, 1 mM DTT, and 1× protease inhibitor cocktail (Roche)), resuspended in 3 ml lysis buffer (50 mM Tris pH 8.0, 300 mM NaCl, 10 mM imidazole, 1 mM DTT, and 1x protease inhibitor cocktail (Roche)) and lysed by sonication. The supernatant was cleared by centrifugation at 21.000 x *g* and 4 °C for 5 min and one aliquot was taken as reference for later enrichment factor calculation. The rest of the supernatant was incubated with 1 ml Ni-NTA bead suspension (Qiagen) on a rotary shaker at 4 °C for 1 h. The suspension was then loaded into a column, beads were washed twice with wash buffer and then eluted with three times with 0.5 ml of elution buffer (50 mM Tris pH 8.0, 300 mM NaCl, 250 mM imidazole, 1 mM DTT, and 1× protease inhibitor cocktail (Roche)). The pulldown procedure was repeated in the presence of 50 µM C4-HSL in all buffers.

Analysis followed published protocols[63]. Briefly, protein extracts (full extract and pooled Ni-NTA eluates) were fractionated by SDS-PAGE, lanes were cut into three parts and then incubated with trypsin (Promega) at 37 °C overnight. Peptide samples were separated with a nano-flow ultra-high pressure liquid chromatography system (RSLC, ThermoFisher Scientific) equipped with a trapping column (3 µm C18 particle, 2 cm length, 75 µm ID, Acclaim PepMap, ThermoFisher Scientific) and a 50 cm long separation column (2 µm C18 particle, 75 µm ID, Acclaim PepMap, ThermoFisher Scientific). Peptide mixtures were injected, enriched and desalted on the trapping column at a flow rate of 6 µL/min with 0.1% TFA for 5 min. The trapping column was switched online with the separating column and peptides were eluted with a multi-step binary gradient: linear gradient of buffer B (80% ACN, 0.1% formic acid) in buffer A (0.1% formic acid) from 4 to 25% in 30 min, 25 to 50% in 10 min, 50 to 90% in 5 min and 10 min at 90% B. The column was reconditioned to 4% B in 15 min. The flow rate was 250 nL/min and the column temperature was set to 45 °C. The RSLC system was coupled online *via* a Nano Spray Flex Ion Source II (ThermoFisher Scientific) to an LTQ-Orbitrap Velos mass spectrometer. Metal-coated fused-silica emitters (SilicaTip, 10 µm i.d., New Objectives) and a voltage of 1.3 kV were used for the electrospray. Overview scans were acquired at

a resolution of 60 k in a mass range of m/z 300–1600 in the orbitrap analyzer and stored in profile mode. The top-10 most intensive ions of charges two or three and a minimum intensity of 2000 counts were selected for CID fragmentation with a normalized collision energy of 38.0, an activation time of 10 ms and an activation Q of 0.250 in the LTQ. Fragment ion mass spectra were recorded in the LTQ at normal scan rate and stored as centroid m/z value and intensity pairs. Active exclusion was activated so that ions fragmented once were excluded from further fragmentation for 70 s within a mass window of 10 ppm of the specific m/z value.

The generated raw files were searched against an in-house data base of *P. aeruginosa* (Häussler group) using MaxQuant software[64] (Version 1.5.3.8) with default settings including mass accuracy of 4.5 ppm for precursor masses and 0.5 Da for ion-trap generated MS-2 fragments. Deamidation at Asp and Gln was set as additional modification.

Proteins were stated as identified by an FDR < 0.01 on peptide and protein level.

Normalization employed the LFQ algorithm[65]. Enrichment was calculated by dividing the LFQ intensities of Ni-NTA eluates by values for the full extract. Non-specifically enriched proteins were eliminated by subtraction of the control strain. Experiments were performed in biological triplicates, and the significance of enrichment was calculated by Student's *t*-test.

## Microscalar thermophoresis (MST)

150 µM of non-tagged RhlR protein purified in the presence of mBTL was first buffer-exchanged into labeling buffer (50 mM Tris pH 7.5, 500 mM NaCl, 5% glycerol, 50 µM mBTL) using Zeba Spin desalting columns. RhlR was then mixed with a 1.5-fold molar excess of ATTO488-maleimid and incubated on ice for 2 h in the dark. Conjugated protein was separated from unreacted dye by gel filtration using a NAP-5 desalting column (Cytiva) and buffer exchanged to RhlR buffer A (50 mM Tris pH 8.0, 500 mM NaCl, 5% glycerol, 50 µM mBTL). The purified protein was then aliquoted and flash frozen in liquid nitrogen for storage at −80 °C until further use.

Before every measurement, a single aliquot of RhlR and PqsE was thawed and briefly centrifuged (10 min, $17.000 \times g$). PqsE was diluted 1:1 into assay buffer (50 mM Tris pH 8.0, 300 mM NaCl, 5% glycerol, 50 µM mBTL, 0.05% Tween) starting from 10 µM and mixed with an equal volume of RhlR-ATTO488 (final concentration: 20 nM). Samples were then incubated for 10 min at RT, loaded into standard-treated capillaries and measured using a Monolith NT.155 (NanoTemper Technologies GmbH) at 25 °C. For the evaluation, samples with irregular MST traces (notable aggregates) were omitted.

## Determination of protein melting points

Protein melting points were determined with a Tycho NT.6 instrument (NanoTemper Technologies GmbH). Proteins dissolved to 0.5–3 mg/ml in the indicated buffer were loaded into glass capillaries and the melting temperature was obtained as the inflection point of the fluorescence ratio at 350 and 330 nm with respect to temperature.

## In-silico design of stabilized RhlR

The "Protein Repair One-Stop Shop" (PROSS) web server[43] was used to design a stability-optimized variant of RhlR. The design was based on a RhlR in-silico model generated with Phyre2[66]. PROSS suggested 75 single point mutations, and the corresponding gene (RhlR-P75) was synthesized by GenScript Biotech Corporation and cloned into plasmid vector pVP008. Internal *Pst*I and *Kpn*I restriction sites were used to replace the mutated DNA-binding domain of RhlR-P75 with the native RhlR sequence followed by QuikChange mutagenesis, yielding RhlR-P61. Both variants were obtained as recombinant proteins by heterologous expression in *E. coli* using the methods described above.

## Protein crystallography

Structure determination by protein crystallography followed standard protocols. Briefly, initial crystallization conditions were identified with automated procedures using the sitting drop vapor diffusion method, and then optimized by grid screening. All crystallization experiments were performed at room temperature. Final crystallization conditions are summarized in Supplementary Table 6. Diffraction data of flash-cooled crystals were collected at 100 K on beamline P11 of the PETRAIII synchrotron and reduced with XDS[67] or STARANISO[68], using AIMLESS from the CCP4 software suite[69] for scaling. One crystal was used per reported structure. Phasing was achieved by molecular replacement in PHASER[70], and refinement involved alternating rounds of manual adjustments in COOT[71] and minimization with phenix.refine of the PHENIX software suite[72]. Data collection and refinement statistics are listed in Tables S7 & S8. Figures have been prepared with PyMOL[73].

## Peptide spot arrays

Peptide arrays were synthesized as previously described[40]. In brief, 21mer peptides with an offset of 3 amino acids were synthesized via Fmoc-synthesis on a solid cellulose support modified with β-alanine, using a customized Intavis Multipep synthesizer. Peptide arrays were blocked with 2% casein in TBS pH 8.0 + 0.1% Tween 20 (TBST) overnight at room temperature, followed by incubation with His$_6$-tagged PqsE at a concentration of 5 µg/ml for 24 h. Cellulose sheets were than washed 3 times with TBST for 5 min before incubation with primary AP-conjugated mouse anti-His$_6$ antibody (GeneTex, GTX44021, f.c. 1:1.000), diluted 1:1000 in blocking buffer for 1.5 h. Sheets were then washed twice with TBST for 5 min, followed by two washing steps in CBS (137 mM NaCl; 2 mM KCl; 50 mM citric acid monohydrate; pH 7.0). Finally, bound antibodies were visualized with CBS containing 50 mM MgCl$_2$, 7.4 mM BCIP (5-bromo-4-chloro-3-indolyl phosphate) in DMF and 7.2 mM MTT in 70% DMF + 30% H$_2$O after incubation for 30 min.

## Reporting summary

Further information on research design is available in the Nature Portfolio Reporting Summary linked to this article.

## Data availability

The data that support this study are available from the corresponding author upon request. Coordinates and diffraction data have been deposited in the Protein Data Bank[39] with accession codes 7R3F (PqsE mutant E187R), 7R3E (PqsE-XTEN30-RhlR fusion protein in complex with mBTL), 8B4A (PqsE/RhlR:C4-HSL complex), 7R3J (PqsE/RhlR:mBTL complex), 7R3G (RhlR-P75 in complex with mBTL), 7R3H (RhlR-P75 in complex with C4-HSL) and 7R3I (RhlR-P61 in complex with mBTL).

Other protein structures used for analysis were obtained from the Protein Data Bank with accession codes 2UV0 (Structure of the *P. aeruginosa* LasR ligand-binding domain bound to its autoinducer), 2Q0I (Structure of *Pseudomonas* Quinolone Signal Response Protein PqsE) and 6MWL (LasR LBD:mBTL complex). Mass spectrometry proteomics data have been deposited to the ProteomeXchange Consortium via the PRIDE[74] partner repository with data set identifier PXD038000. Source data are provided with this paper.

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

## Acknowledgements

We are grateful to the staff of beamline P11 at the PETRAIII synchrotron (DESY campus Hamburg, Germany) for letting us use their facilities. Peer Lukat is acknowledged for help with data collection. S.R.B. received a scholarship by the Life Sciences Foundation and was supported by the HZI International Graduate School for Infection Research (GS-FIRE). S.He. is a member of the PROCOMPAS graduate school supported by the Deutsche Forschungsgemeinschaft (281361126/GRK2223).

## Author contributions

R.S.B., S.He., F.W., S.Sch., S.z.L, M.K., S.-K. H., M.v.H, L.J., M.B., N.O.G., S.Häu., J.K., A.P. and W.B. designed and/or performed experiments. R.S.B., S.He., F.W., S.Sch., S.-K.H., M.v.H, L.J., M.B., N.O.G., S.Häu., J.K., A.P. and W.B. analyzed data. D.L. and S.St. generated critical reagents. W.B. wrote the manuscript.

## Funding

## Competing interests

The authors declare no competing interests.
