## [Peer Review File · Nature Communications]

Reviewers' Comments:

Reviewer #1:

Remarks to the Author:

The work by Borgert et al describes the detailed structural characterization of the complex formed by the *Pseudomonas aeruginosa* QS-transcriptional regulator RhIR and the PqsE protein and show that RhIR is stabilized by this interaction. Furthermore, the authors show through extensive mutagenesis of RhIR that PqsE causes a conformational change that stabilizes this QS-transcriptional regulator and renders a RhIR conformation that activates the expression of genes involved in pyocyanin production. RhIR mutant (RhIR-P61) was selected as a heat stable derivative that has no changes in its DNA-binding site, and is PqsE independent for the production of pyocyanin.

I consider that the evidence that is presented in this article shows that the formation of the RhIR-PqsE complex represents the main mechanism by which the Pqs QS-system modulates RhIR activity in the synthesis of pyocyanin.

Therefore, this work represents a major break through in the understanding of the regulation of *P. aeruginosa* QS-dependent virulence factors production, and it might impact the development of therapeutic alternatives to the use of antibiotics that target the inhibition of this bacterial pathogen, as described in the manuscript. Therefore I recommend its publication in *Nature Communications* after minor modifications.

My main concerns and suggestions are:

1. In line 120 the authors state that "RhIR is intrinsically unstable", but I consider that there are no data in this article documenting the intrinsic instability of RhIR and that there is reported evidence, and also data presented in this work, that shows that this is not the case.

In Fig. 2a, the presence of RhIR is apparent in the soluble fraction even in the absence of C4-HSL or mBTL. Furthermore it has been reported that RhIR is able to bind to rhIA "las box" in the absence of C4-HSL both in *P. aeruginosa* as in *E. coli* (doi: 10.1128/JB.185.20.5976-5983.2003; <https://doi.org/10.1099/mic.0.050161-0>; doi: 10.1111/1574-6968.12505), so this protein is not intrinsically unstable in the absence of its cognate AI, as has been reported for other members of the LuxR-family of transcriptional regulators as TraR, LuxR and LasR. In addition, pqsE mutants do not produce PYO, but are able to produce rhamnolipids, so in the absence of PqsE RhIR/C4-HSL is functional for the activating the transcription of the rhlAB operon.

Even though PqsE stabilizes RhIR as is extensively shown in this work, RhIR cannot be considered as intrinsically unstable. Maybe it can be stated in line 120 that PqsE participates in the QS response through modulating RhIR activity as the reason for investigating the RhIR-PqsE complex.

2. I consider that defining the moonlight activity of PqsE as "chaperon-like" constricts the biological significance that its ability to forms complexes with RhIR might have. It might be that PqsE has an additional role besides its chaperon-like activity defined as increasing the stability of RhIR, since RhIR-PqsE complexes affect specifically the expression of genes involved in pyocyanin production and have a slight effect in rhamnolipids production (as shown in your reference 32). It is thus possible that RhIR when complexed with PqsE might have a slightly modified DNA binding specificity, since for example the phzA2 promoter region does not have a distinct "las box" where RhIR can bind DNA but its expression is activated by RhIR/C4-HSL/PqsE. These are speculations and experimental evidence to support these conjectures is needed. I suggest that it is mentioned in the discussion section that the molecular mechanism of the transcriptional activation of the RhIR/C4-HSL/PqsE complex needs to be further studied.

3. The complexation of PqsE with RhIR might not only increase its stability but it might also protect RhIR from oxidation by pyocyanin or by the reactive oxygen species (ROS) produced by this phenazine. This possibility could be experimentally validated or at least this point should be mentioned in the article.

Minor comments:

1. Instead of the WHO link (reference 10) cite the following article:

Shrivastava, S.R.; Shrivastava, P.S.; Ramasamy, J. Responding to the challenge of antibiotic resistance: World Health Organization. *J. Res. Med. Sci.* 2018, 23, 21.

2. Define LBD as "ligand binding domain" the first time you use this term (line 176) and use LBD in line 184.

Reviewer #2:

Remarks to the Author:

The authors reported that PqsE stabilizes and increases the soluble form of RhIR via chaperone-like activity mediated by direct complex formation with RhIR. They solved the crystal structure of the complex (PqsE-XTEN30-RhIR:mBTL) in a sophisticated way using a linker peptide, and also did the structure of PqsE-RhIR:mBTL without an additional linker at more high resolution. In addition, the RhIR derivative proteins with higher stability (RhIR-P75 and -P61) were also designed, and then their complex structures with mBTL were also determined. These studies seem to be the structural support of the recently published paper (The PqsE-RhIR interaction regulates RhIR DNA binding to control virulence factor production in *Pseudomonas aeruginosa*. *Microbiol Spectr* 2022, 10, 1, e0210821). However, there are several points that should be addressed to clarify the manuscript.

Major.

1. Page-2, 40th – The interaction of PqsE and RhIR does not seem to be weak, since the 2:2 complex is identified in SEC-MALS analysis. Therefore, the measurements of the K_d values of RhIR (or RhIR:mBTL) at least for the native PqsE and additionally for the monomeric PqsE (E187R) would be necessary to clarify the *in vivo* meaning of this complex in terms of protein concentration.

2. Page-5, 113th & page-12, 263th – I am not sure whether the “unprecedented chaperone-like activity” is true or not. As I know, there are several cases in which a protein is stabilized by the presence of another interacting protein; e.g. the soluble expression of LsrK protein is dependent on the presence of Hpr (*Sci Adv* 2018, 4, 6, eaar7063).

In a similar line, it should be clearly described if the soluble fraction of RhIR was only increased by the presence of PqsE and mBTL since the overall amount of RhIR of whole-cell lysates does not change in Fig. 2a.

3. The 3D structure of RhIR is the first report, and thus it would be necessary to address the antagonistic effect of mBTL compared to BTL (*PNAS* 2013, 110, 44, 17981-6) based on the complex structure.

4. People believe that (i) an agonist-binding releases the repressor protein from the cognitive promoter region, but (ii) an antagonist-binding stabilizes the repressor-promoter complex inhibiting the transcription. It has recently been reported that PqsE-RhIR:mBTL has a higher binding affinity for the promoter DNA than RhIR:mBTL does (*Microbiol Spectr* 2022, 10, 1, e0210821). I am not sure if the case of RhIR for the agonist (C4-HSL) and antagonist (mBTL) agrees with the general mechanism of repressor proteins mentioned above. If it is true, it would be much easier for readers to understand the difficulty to obtain RhIR:C4-HSL compared to RhIR:mBTL. Description of the known RhIR mechanism of the promoter recognition would be useful for the general reader's expanding discussion based on the author's structural data.

5. The symmetry breaking in the 2:2 complex of PqsE and RhIR:mBTL is normal since a structural mismatch between the two dimeric RhIR and PqsE is inevitable. The mutations in the contact area 2 of PqsE do not seem to have any serious effect (Extended Data Fig. 3c, right panel; the mutations of RhIR may cause a structural defect in RhIR itself). Is there any interpretation for this?

6. It would be useful to discuss the mechanism of the previously reported RhIR mutant, such as RhIR* (RhIR Y64F W68F V133F triple mutant; *PLoS Pathog* 2019, 15, 6, e1007820), based on the solved crystal structures of PqsE-RhIR:mBTL, and various RhIR:mBTL.

Minor.

1. It would be better to show the chemical structure of mBTL in Fig. 4b.

2. The Extended Data Fig. 5 should be mentioned in the result section first since it is experimental data.

Reviewer #3:

Remarks to the Author:

In this manuscript, Borgert and Henke et al. report on the *rhl* circuit in *P. aeruginosa*, playing the leading role in late and chronic infections. They elucidate the mechanism of the transcription factor RhIR, which depends on PqsE, a thioesterase. They show that PqsE and RhIR form a weakly interacting protein complex together with C4-HSL and they used different structural methods to

elucidate this interaction. They identify a chaperone-like function of PqsE, enhancing the stability of RhIR via protein complex formation.

To validate this, some of the major techniques involved were crystallography and mutation screening, but also one peptide array experiment. This reviewer was asked to focus on the peptide array approach.

The overall peptide array approach appears somewhat sloppily presented, several details are missing. The array data is only discussed in two sections of the whole manuscript. Apparently, two array copies (for +PqsE experiment and -PqsE control), covering the RhIR sequence as 21-mer peptides with 18 AA overlap, were synthesized on a cellulose membrane via the SPOT synthesis approach. Neither the source of the RhIR sequence, nor the synthesized peptides were reported in the manuscript or supporting information. One of the arrays was incubated with His6-tagged PqsE and stained with an AP-conjugated anti-His6 antibody and BCIP, the other (control) only with AP-conjugated anti-His6 antibody and BCIP.

Binding events (experiment vs. control) were very weak, as probably expected, but the background signals are much stronger than the actual considered binding. Therefore, interpretation of signals is difficult. For example: Why was peptide AA37-AA57 not considered as a binding event, while AA136-AA156 was considered (which has a quite similar signal intensity). The interpretation of binding appears more like an educated guess, which stems from the crystallography data. The strongest signals, AA52-AA78, do not correlate with the crystallography data and they are not discussed at all in the manuscript. Concluding, the reported array data is not really convincing and currently raises more questions than supporting the main claims and crystallography data. Thus, the array data could be removed from the manuscript or shifted to the supporting information.

Main concerns regarding the peptide array approach:

1. Source sequence of the original RhIR protein is not mentioned.
2. Sequences of the peptide spots on the array are nowhere to be found. Please at least add this to the supporting information.
3. Negative control without PqsE shows many strong signals already. There is no discussion on why this might be. The authors should give at least a hypothesis on this, since the interpreted signals are not very convincing.
4. Interpretation of binding events appears as an educated guess from the crystallography data. Why was peptide AA37-AA57 not considered as a binding event, while AA136-AA156 was considered (which has a quite similar signal intensity)?
5. The authors used AP-conjugated anti-His6 antibody from GeneTex (GTX44021). Did the authors realize that this has been discontinued by GeneTex due to quality issues?
<https://www.genetex.com/Product/Detail/6X-His-tag-antibody-AD1-1-10-AP/GTX44021>
The authors should contact the company and get more details on whether their antibody LOT had quality issues. Did the authors repeat their experiments with another staining method or a different AP-anti-His6 antibody?
6. If the authors want to keep the array data in the main text, maybe the authors should validate their array data by another peptide assay (ELISA?).
7. In general, the reporting summary does not contain any specific information on the employed antibodies. Please provide product/catalogue and LOT number.

POINT-BY-POINT REPLY TO REVIEWER COMMENTS (REPLIES IN BLUE)

Reviewer #1 (Remarks to the Author):

The work by Borgert et al describes the detailed structural characterization of the complex formed by the *Pseudomonas aeruginosa* QS-transcriptional regulator RhIR and the PqsE protein and show that RhIR is stabilized by this interaction. Furthermore, the authors show through extensive mutagenesis of RhIR that PqsE causes a conformational change that stabilizes this QS-transcriptional regulator and renders a RhIR conformation that activates the expression of genes involved in pyocyanin production. RhIR mutant (RhIR-P61) was selected as a heat stable derivative that has no changes in its DNA-binding site, and is PqsE independent for the production of pyocyanin.

I consider that the evidence that is presented in this article shows that the formation of the RhIR-PqsE complex represents the main mechanism by which the Pqs QS-system modulates RhIR activity in the synthesis of pyocyanin.

Therefore, this work represents a major break through in the understanding of the regulation of *P. aeruginosa* QS-dependent virulence factors production, and it might impact the development of therapeutic alternatives to the use of antibiotics that target the inhibition of this bacterial pathogen, as described in the manuscript. Therefore I recommend its publication in Nature Communications after minor modifications.

My main concerns and suggestions are:

1. In line 120 the authors state that "RhIR is intrinsically unstable", but I consider that there are no data in this article documenting the intrinsic instability of RhIR and that there is reported evidence, and also data presented in this work, that shows that this is not the case.

In Fig. 2a, the presence of RhIR is apparent in the soluble fraction even in the absence of C4-HSL or mBTL. Furthermore it has been reported that RhIR is able to bind to rhlA "las box" in the absence of C4-HSL both in *P. aeruginosa* as in *E. coli* (doi: 10.1128/JB.185.20.5976-5983.2003; <https://doi.org/10.1099/mic.0.050161-0>; doi: 10.1111/1574-6968.12505), so this protein is not intrinsically unstable in the absence of its cognate AI, as has been reported for other members of the LuxR-family of transcriptional regulators as TraR, LuxR and LasR. In addition, pqsE mutants do not produce PYO, but are able to produce rhamnolipids, so in the absence of PqsE RhIR/C4-HSL is functional for the activating the transcription of the rhlAB operon.

Even though PqsE stabilizes RhIR as is extensively shown in this work, RhIR cannot be considered as intrinsically unstable. Maybe it can be stated in line 120 that PqsE participates in the QS response through modulating RhIR activity as the reason for investigating the RhIR-PqsE complex.

We thank the referee for commenting so positively on our work.

Regarding the use of the term "intrinsic instability of RhIR", we agree that our original wording was probably too biased towards our research focus in structural biology, where "stability" normally implies timescales of days or weeks. We have therefore reworded the respective sections of the manuscript accordingly and have exchanged "stability" for "solubility" wherever it seemed appropriate.

However, we are still convinced that RhIR is much less stable than most other proteins of mesophilic bacteria, and that C4-HSL and PqsE significantly enhance the amounts of properly folded RhIR when the protein is expressed. In preparation of the revised manuscript, we have therefore conducted a series of experiments to demonstrate this better. Optimized western blotting was used to assess the overexpression of RhIR in *E. coli* in the presence of increasing amounts of C4-HSL and in the presence or absence of PqsE (new Fig. 2a). Proteomic analysis was used to assess the amount of total and soluble RhIR in *P. aeruginosa* and various deletion variants (new Supplementary Fig. 1b). In addition, we have used SDS-PAGE to observe time-dependent stability of purified recombinant proteins (new Fig. 2c). Stability was inferred from the amount of protein in the soluble fraction of the respective samples. Together, these new experiments led to the following major findings:

- *E. coli* produces only minute amounts of soluble RhIR in the absence of C4-HSL, and PqsE does not improve solubility when C4-HSL is lacking;
- There is a concentration window extending up to at least 50 μ M C4-HSL in which PqsE increases the amount of soluble (= stable) RhIR significantly. This is particularly apparent at C4-HSL concentrations up to 10 μ M;

- The PqsE/RhIR complex purified in the presence of C4-HSL loses RhIR almost completely after 4 hours when incubated at 37 °C, whereas PqsE did not change significantly within 24 hours (new Fig. 2c, left panel);
- mBTL has previously been demonstrated to increase the yield of soluble RhIR with respect to C4-HSL (Fig. 2D of O'Loughlin et al., PNAS (2013) 110:17981-6), however, mBTL still fails to increase the long-term stability of RhIR as can be seen from the full precipitation of RhIR purified in the presence of mBTL (new Fig. 2c, right panel);

Regarding the non-essentiality of C4-HSL for DNA-binding of RhIR as referenced in the three publications the referee mentions, direct evidence for this is provided by EMSA as shown in Fig. 2 of the report by Medina et al. (J. Bacteriol. (2003) 185:5976-83). However, these EMSAs have been obtained with lysate from *E. coli* very strongly overproducing RhIR (the authors report that 5% of the total cell protein was RhIR), but it is unclear how this lysate has been treated and how much soluble RhIR it may have contained. Further, the same report shows significantly enhanced production of RhIR in the presence of C4-HSL (Fig. 5B of Medina et al.), which is similar to our findings reported in the new Fig. 2a and Supplementary Fig. 1b. Thus, the work the referee refers to is not contradictory to our findings.

The remark about rhamnolipid production in the absence of PqsE is addressed below.

2. I consider that defining the moonlight activity of PqsE as “chaperon-like” constricts the biological significance that its ability to form complexes with RhIR might have. It might be that PqsE has an additional role besides its chaperon-like activity defined as increasing the stability of RhIR, since RhIR-PqsE complexes affect specifically the expression of genes involved in pyocyanin production and have a slight effect in rhamnolipids production (as shown in your reference 32). It is thus possible that RhIR when complexed with PqsE might have a slightly modified DNA binding specificity, since for example the *phzA2* promoter region does not have a distinct “las box” where RhIR can bind DNA but its expression is activated by RhIR/C4-HSL/PqsE. These are speculations and experimental evidence to support these conjectures is needed. I suggest that it is mentioned in the discussion section that the molecular mechanism of the transcriptional activation of the RhIR/C4-HSL/PqsE complex needs to be further studied.

This point is well taken. The referee correctly reminds us that not all RhIR-regulated genes show the same dependence on PqsE and that there are genes that do not require PqsE at all to be up- or downregulated by RhIR, with rhamnolipid biosynthesis being a prominent example. This has probably been most comprehensively studied in a very recent report by Letizia et al. (Microbiol. Spectr. (2022), 10:e0096122), which is now included in our revised manuscript (new reference (16)). First, this work demonstrates that C4-HSL is absolutely required for RhIR-dependent regulation, which reflects findings in our new Fig. 2a. Second, the RhIR-regulon subdivides into three categories, namely (1) genes that require PqsE to be induced at all, (2) genes that do not require PqsE but are more strongly regulated when PqsE is present, and (3) RhIR-regulated genes that do not respond to PqsE. A similar classification was also observed by Mukherjee et al. (reference (22), PNAS (2018), 115:E9411-8), and these categories have now been included in the updated Fig. 1 of the revised manuscript.

While this could hint at different DNA binding modes that depend on allosteric structural effects triggered by PqsE, it is also reasonable that these differences correlate to the DNA binding affinity of RhIR, which in turn would lead to differences in sensing RhIR concentration levels. The finding by Letizia et al. (16) that a large number of genes in the second group (not requiring PqsE for induction but being more strongly regulated in its presence) seems to support such a concentration-sensing mechanism. However, reviewer #1 is correct that our current data do not allow us to rule out allosteric effects by PqsE, and we have emphasized this by slightly expanding the respective statement in the discussion section (now at lines 337-341; lines 267-269 of the original manuscript) as well as by rewording the title of the manuscript.

3. The complexation of PqsE with RhIR might not only increase its stability but it might also protect RhIR from oxidation by pyocyanin or by the reactive oxygen species (ROS) produced by this phenazine. This possibility could be experimentally validated or at least this point should be mentioned in the article.

The point made by the referee raises the important question about how RhIR is inactivated in general, and the idea that oxidation by pyocyanin contributes to this is interesting. RhIR contains three cysteine residues not involved in disulfide bridge formation, which could indeed hint at oxidation-dependent regulation. However, pyocyanin is only one of several secreted redox-active compounds that *P.*

aeruginosa produces. A thorough analysis would require a significantly broader experimental approach that seems beyond the scope of our manuscript. We have added a respective statement in lines 168-171 of the revised manuscript.

Minor comments:

1. Instead of the WHO link (reference 10) cite the following article:

Shrivastava, S.R.; Shrivastava, P.S.; Ramasamy, J. Responding to the challenge of antibiotic resistance: World Health Organization. *J. Res. Med. Sci.* 2018, 23, 21.

We thank the referee for suggesting an alternative to the WHO website; the link to a specific page is indeed not ideal and has in fact expired in the meantime. However, the suggested reference is a comment rather than an article, and it does not mention *P. aeruginosa* specifically. We have now included an alternative publication by the WHO Pathogens Priority List Working Group, who had published their work in *Lancet Infect. Dis.* (2018) 18:318-27.

2. Define LBD as "ligand binding domain" the first time you use this term (line 176) and use LBD in line 184.

The definition has been included.

Reviewer #2 (Remarks to the Author):

The authors reported that PqsE stabilizes and increases the soluble form of RhIR via chaperone-like activity mediated by direct complex formation with RhIR. They solved the crystal structure of the complex (PqsE-XTEN30-RhIR:mBTL) in a sophisticated way using a linker peptide, and also did the structure of PqsE-RhIR:mBTL without an additional linker at more high resolution. In addition, the RhIR derivative proteins with higher stability (RhIR-P75 and -P61) were also designed, and then their complex structures with mBTL were also determined. These studies seem to be the structural support of the recently published paper (The PqsE-RhIR interaction regulates RhIR DNA binding to control virulence factor production in *Pseudomonas aeruginosa*. *Microbiol Spectr* 2022, 10, 1, e0210821). However, there are several points that should be addressed to clarify the manuscript.

Major.

1. Page-2, 40th – The interaction of PqsE and RhIR does not seem to be weak, since the 2:2 complex is identified in SEC-MALS analysis. Therefore, the measurements of the K_D values of RhIR (or RhIR:mBTL) at least for the native PqsE and additionally for the monomeric PqsE (E187R) would be necessary to clarify the in vivo meaning of this complex in terms of protein concentration.

We thank the referee for this important remark and also for encouraging us to try quantifying the interaction between PqsE and RhIR. We have employed microscalar thermophoresis (MST) to measure the affinity of PqsE-wildtype and the of PqsE-E187R mutant to RhIR purified in the presence of mBTL. The K_D value for PqsE-wildtype was found to be in the 150 nM range (new Supplementary Fig. 3d, left panel), showing that the interaction between both proteins is indeed not weak and suggesting that the instability of the complex arises from RhIRs tendency to precipitate over time (shown in new Fig. 2c). Importantly, the E187R variant of PqsE showed significantly lower affinity to RhIR:mBTL ($K_D \approx 14 \mu\text{M}$; new Supplementary Fig. 3d, right panel), corroborating our hypothesis that dimerization is an important requirement for complex formation with RhIR.

2. Page-5, 113th & page-12, 263th – I am not sure whether the “unprecedented chaperone-like activity” is true or not. As I know, there are several cases in which a protein is stabilized by the presence of another interacting protein; e.g. the soluble expression of LsrK protein is dependent on the presence of Hpr (*Sci Adv* 2018, 4, 6, eaar7063).

The point is well taken. We have reworded the respective section to indicate that this is, to our knowledge, the first example of a LuxR-type transcription factor to be stabilized by interaction with another protein rather than the first general example of such a stabilizing complex (line 324-325).

In a similar line, it should be clearly described if the soluble fraction of RhIR was only increased by the presence of PqsE and mBTL since the overall amount of RhIR of whole-cell lysates does not change in Fig. 2a.

The respective section has been completely rewritten, following also the criticism of reviewers #1 and #3. Importantly, we have performed new experiments to better show the requirement of C4-HSL and PqsE for enhancing the solubility of RhIR in *E. coli* overproducing the protein (new Fig. 2a) and also in *P. aeruginosa* (new Supplementary Fig. 1b). The fact that the total amount of RhIR does not change is stressed in lines 134-136 and shown in Supplementary Fig. 1a/b.

3. The 3D structure of RhIR is the first report, and thus it would be necessary to address the antagonistic effect of mBTL compared to BTL (PNAS 2013, 110, 44, 17981-6) based on the complex structure.

The reviewer raises an important question that our current data do not allow us to answer unequivocally. From a modeling point of view, one may speculate that a halogen atom in the para instead of the meta position in the respective compound (the indicated paper reports data for a compound termed “CTL” here) clashes with the protein:

Reviewer-only figure: Surface around mBTL in the PqsE/RhIR:mBTL crystal structure (left). This structure was then used to construct a simple model of CTL posed identically (right). Note that the para-chloro-phenolate group is expected to clash with the protein if not adopting a different conformation.

However, since no affinity data for CTL or mBTL have been reported and since we do not have access to CTL to perform additional experiments with e.g. one of our stabilized RhIR variants, we would like to abstain from discussing this point in our manuscript.

With respect to the antagonistic activity of mBTL in wildtype *P. aeruginosa*, we want to re-emphasize though that in contrast to the data reported in the indicated paper (O’Laughlin et al., PNAS (2013), 110:17981-6), our experiments indicate that mBTL acts agonistically to similar levels as externally added C4-HSL in a RhII deletion mutant that cannot produce C4-HSL. As discussed in our manuscript, we attribute this to mBTLs stabilizing effect on RhIR (new Supplementary Fig. 1c, ED Fig. 5 of the original manuscript).

4. People believe that (i) an agonist-binding releases the repressor protein from the cognitive promoter region, but (ii) an antagonist-binding stabilizes the repressor-promoter complex inhibiting the transcription. It has recently been reported that PqsE-RhIR:mBTL has a higher binding affinity for the promoter DNA than RhIR:mBTL does (Microbiol Spectr 2022, 10, 1, e0210821). I am not sure if the case of RhIR for the agonist (C4-HSL) and antagonist (mBTL) agrees with the general mechanism of repressor proteins mentioned above. If it is true, it would be much easier for readers to understand the difficulty to obtain RhIR:C4-HSL compared to RhIR:mBTL. Description of the known RhIR mechanism of the promoter recognition would be useful for the general reader’s expanding discussion based on the author’s structural data.

We thank the referee for this insightful comment. LuxR-type transcription factors (TFs) can indeed act as transcriptional activators or repressors, and the mechanisms that the referee refers to here, i.e. derepression by release from the promoter upon agonist/autoinducer binding as well as repression via enhanced affinity towards DNA after antagonist/autoinducer binding, have both been described. However, in most cases, LuxR-type TFs are known to activate transcription by association with the promoter after binding of the agonist/autoinducer. The situation is further complicated by the fact that LuxR-type TFs can simultaneously act as activators for one set of genes and as repressors for another. In addition to enhancing affinity towards the DNA allosterically, ligand binding is in some LuxR-TFs (a group to which RhIR seems to belong) also known to be required for folding and/or dimerization of the TF as a prerequisite for being able to recognize the promoter sequence. This has been reviewed by Churchill and Chen (Chem. Rev. (2011) 111:68-85), which is now cited as blue reference (25).

In the case of RhIR, complexity is further expanded by the interplay with PqsE. In a report published while our manuscript was under review (Letizia et al., 2022) and in earlier work (Mukherjee et al., 2018), colleagues showed that this interplay leads to a regulon segregated into three subgroups, i.e. one that does not require PqsE, a second one that is enhanced by PqsE and a third that strictly depends on PqsE. A specific promoter sequence seems to characterize each of these groups, and importantly, all three of them contain up- as well as down-regulated genes at the same time (Letizia et al., 2022). We have included this information in the revised Fig. 1.

Since our data indicate that RhIR requires C4-HSL and PqsE for folding, i.e. to attain a DNA-binding-compatible conformation, and since the synthetic ligand mBTL is known to stabilize RhIR while acting both antagonistically in wildtype *P. aeruginosa* (O'Loughlin et al., 2013) but agonistically in a C4-HSL-deficient mutant (Supplementary Fig. 1c), we feel that it is currently not possible to make a conclusive statement about activation and repression mechanisms for RhIR-regulated genes.

The referee correctly indicates that DNA-binding affinities may lead towards better insight. However, the required quantitative affinity data are difficult to obtain: ligand-free or C4-HSL-bound RhIR for comparison with mBTL-bound RhIR cannot be isolated, and results of affinity measurements may ultimately be misleading because they will be influenced by the amount of folded RhIR in specimen used for such experiments. The level of folded RhIR is likely to be different in samples containing mBTL, C4-HSL and PqsE. Such studies are important but seem beyond the scope of our current manuscript. We will try to address this aspect in future studies.

5. The symmetry breaking in the 2:2 complex of PqsE and RhIR:mBTL is normal since a structural mismatch between the two dimeric RhIR and PqsE is inevitable. The mutations in the contact area 2 of PqsE do not seem to have any serious effect (Extended Data Fig. 3c, right panel; the mutations of RhIR may cause a structural defect in RhIR itself). Is there any interpretation for this?

The statement about the expected symmetry break is correct. However, we were surprised when we first saw it in our crystal structures, since from an evolutionary viewpoint, symmetrical interfaces have the advantage that beneficial mutations lead to a multifold stabilization of the resulting complex, this probably being one of the reasons why so many proteins are homooligomers. On top of this, because of the short lifetime of the purified complex we (falsely) concluded that the interaction between both proteins is relatively weak and therefore prone to be tweaked by crystal forces. However, the new K_d measurements that the referee encouraged us to perform (see remark 1) have convinced us that there is considerable affinity between both proteins and that the symmetry distortion is not a crystallization artefact. We have rephrased the respective section of the manuscript.

With respect to PqsE mutations that affect interface 2 alone, we assume that the respective interactions do not contribute significantly to the stability of the PqsE/RhIR complex. Further interpretation does not seem possible at the present stage.

With respect to the mutations introduced in RhIR, we can indeed not exclude that they may harm the structural integrity of RhIR. The mutations are at the surface of RhIR, making general destabilization less likely, however. Further, since our experiments show that RhIR requires PqsE to reach significant levels of properly folded protein, at least in the presence of non-excessive amounts of C4-HSL, it is difficult to separate a potential impact on the general integrity of RhIR from interfering with the solubility-enhancing interaction with PqsE when using solubility as a proxy for stability.

6. It would be useful to discuss the mechanism of the previously reported RhIR mutant, such as RhIR* (RhIR Y64F W68F V133F triple mutant; PLoS Pathog 2019, 15, 6, e1007820), based on the solved crystal structures of PqsE-RhIR:mBTL, and various RhIR:mBTL.

The RhIR* mutant reported in the indicated paper by McReady et al. (PLoS Pathog (2019), 15:e1007820) is indeed interesting because it does not seem to require additional stimulation by C4-HSL to upregulate RhIR-dependent genes. These authors observed that RhIR* was significantly more stable than wildtype RhIR, which seems to support our hypothesis that the amount of folded RhIR decides about the transcription level of RhIR-regulated genes. Because of this, we have in fact tried to produce this variant at earlier stages of our work, hoping that it could help us towards structural analysis of the PqsE/RhIR complex. However, in our hands, RhIR* was not easier to handle than wildtype RhIR itself, which then led us to employ PROSS as a more invasive approach to generate the stable variants RhIR-P75 and RhIR-P61 reported in our manuscript.

Without insightful structural data (e.g. apo-RhIR, apo-RhIR* and complexes with C4-HSL), it is difficult to speculate why RhIR* is more stable and if this increased stability is indeed the basis for C4-HSL-independent activity that the authors have reported. The RhIR* mutations Y64F and W68F disrupt two conserved hydrogen bonds to the homoserine lactone moiety of AHL autoinducers bound to LuxR-TFs. The essential mutation that according to the report by McReady and colleagues leads to C4-HSL independent signaling of RhIR* (V133F) is close to the ligand binding pocket and may interfere with the position of the acyl chain of C4-HSL, as simple modeling on the basis of our new PqsE/RhIR:C4-HSL complex reveals:

Reviewer-only figure: Model of a RhIR*/C4-HSL complex based on the new PqsE/RhIR:C4-HSL crystal structure. The side chains of Y64, W68 and V133 (original side chain positions shown in thin lines) have been mutated to phenylalanine in Coot and the best-fitting rotamers have been adopted.

It is conceivable that exchanging the polar residues Y64 and W68 to phenylalanine and replacing V133 with another more space-filling phenylalanine eliminates the need for counterbalancing polar groups in the ligand binding site of RhIR via complex formation with C4-HSL and hence leads to higher stability of RhIR*. However, we deem such explanation as highly speculative and not adding to the focus of our manuscript such that we have not included this aspect in the revised version.

Minor.

1. It would be better to show the chemical structure of mBTL in Fig. 4b.
2. The Extended Data Fig. 5 should be mentioned in the result section first since it is experimental data.

The requested changes have been made. Extended Data Fig. 5 has now been included as panel c of Supplementary Fig. 1 and is first mentioned in the results section.

Reviewer #3 (Remarks to the Author):

In this manuscript, Borgert and Henke et al. report on the rhl circuit in *P. aeruginosa*, playing the leading role in late and chronic infections. They elucidate the mechanism of the transcription factor RhIR, which depends on PqsE, a thioesterase. They show that PqsE and RhIR form a weakly interacting protein complex together with C4-HSL and they used different structural methods to elucidate this interaction. They identify a chaperone-like function of PqsE, enhancing the stability of RhIR via protein complex formation.

To validate this, some of the major techniques involved were crystallography and mutation screening, but also one peptide array experiment. This reviewer was asked to focus on the peptide array approach.

The overall peptide array approach appears somewhat sloppily presented, several details are missing. The array data is only discussed in two sections of the whole manuscript. Apparently, two array copies (for +PqsE experiment and -PqsE control), covering the RhIR sequence as 21-mer peptides with 18 AA overlap, were synthesized on a cellulose membrane via the SPOT synthesis approach. Neither the source of the RhIR sequence, nor the synthesized peptides were reported in the manuscript or supporting information. One of the arrays was incubated with His6-tagged PqsE and stained with an AP-conjugated anti-His6 antibody and BCIP, the other (control) only with AP-conjugated anti-His6 antibody and BCIP.

Binding events (experiment vs. control) were very weak, as probably expected, but the background signals are much stronger than the actual considered binding. Therefore, interpretation of signals is difficult. For example: Why was peptide AA37-AA57 not considered as a binding event, while AA136-AA156 was considered (which has a quite similar signal intensity). The interpretation of binding appears more like an educated guess, which stems from the crystallography data. The strongest signals, AA52-AA78, do not correlate with the crystallography data and they are not discussed at all in the manuscript. Concluding, the reported array data is not really convincing and currently raises more questions than supporting the main claims and crystallography data. Thus, the array data could be removed from the manuscript or shifted to the supporting information.

Main concerns regarding the peptide array approach:

1. Source sequence of the original RhIR protein is not mentioned.
2. Sequences of the peptide spots on the array are nowhere to be found. Please at least add this to the supporting information.
3. Negative control without PqsE shows many strong signals already. There is no discussion on why this might be. The authors should give at least a hypothesis on this, since the interpreted signals are not very convincing.
4. Interpretation of binding events appears as an educated guess from the crystallography data. Why was peptide AA37-AA57 not considered as a binding event, while AA136-AA156 was considered (which has a quite similar signal intensity)?
5. The authors used AP-conjugated anti-His6 antibody from GeneTex (GTX44021). Did the authors realize that this has been discontinued by GeneTex due to quality issues?
<https://www.genetex.com/Product/Detail/6X-His-tag-antibody-AD1-1-10-AP/GTX44021>
The authors should contact the company and get more details on whether their antibody LOT had quality issues. Did the authors repeat their experiments with another staining method or a different AP-anti-His6 antibody?
6. If the authors want to keep the array data in the main text, maybe the authors should validate their array data by another peptide assay (ELISA?).
7. In general, the reporting summary does not contain any specific information on the employed antibodies. Please provide product/catalogue and LOT number.

We need to thank the referee for criticizing the peptide spot arrays reported in our initial manuscript. These experiments were amongst the first ones we did to gain slightly more detailed insight into the interaction of PqsE and RhIR back in 2016/2017. At that time, we could only dream of ever obtaining a crystal structure of the PqsE/RhIR complex due to all of the problems we needed to overcome with respect to the stability of RhIR.

In order to show more convincing data, we invested significant time to produce clearer peptide arrays in preparation of the revised manuscript. Towards this, we synthesized new arrays, tried different antibodies against His₆- and StrepII-tagged versions of PqsE as well as other detection methods such as fluorescently labeled streptavidin to assess the interaction of PqsE with RhIR-derived peptides. Unfortunately, all of these experiments suffered from the same problem, i.e. extremely high background signal that obscured differences to negative controls. This could also not be resolved by optimization of the blocking procedure. We have therefore decided to follow the referee's advice to move the original arrays to the supplementary materials. In addition, because of high background signals in the negative control, we have removed any interpretation of these arrays. We are convinced that this does not impact on the conclusions of our manuscript since the crystal structures and

mutagenesis data that follow in the other parts of the results section reveal higher and unequivocal insight into the interaction between both proteins.

We provide the initially missing information about the original RhIR sequence (RhIR and PqsE of the commonly investigated *P. aeruginosa* strains PAO1 and PA14 are 100% identical) and peptide sequences in Supplementary Tab. 9. Information about the anti-His₆-tag antibody's lot number has been added to Supplementary Tab. 5. With respect to quality issues of this antibody (GTX44021) we contacted both the producer and its German distributor Biozol. We were informed that there were no quality issues at the time of our purchase (summer 2016) and that the antibody has been discontinued due to regulatory/supply issues.

Reviewers' Comments:

Reviewer #1:

Remarks to the Author:

The revised version of the manuscript by Borgert et al., addresses all the issues that I included in my review and provide solid data on which the author based their conclusions. I agree with the change of the title of the article since it more clearly describe the conclusions attained. Among the new data contained in the modified manuscript is the assessment of RhIR solubility and it is clearly shown that it is increased by the complex formed with PqsE and that RhIR interaction with C4-HSL is required independently of PqsE. These results explain the observations reported previously that were interpreted to be due to RhIR stability.

I consider that these results and the detailed structural characterization of RhIR coupled with C4-HSL or with mBTL, as well as of the complex formed with PqsE that requires the dimerization of this enzyme, represent a major contribution to the field of *P. aeruginosa* QS-sensing. Therefore I recommend that this manuscript is accepted for its publication in Nature Communications.

Gloria Soberón-Chávez.

Reviewer #2:

Remarks to the Author:

The authors properly responded to all the questions I raised. Although there is a simple mistake in the reply letter (new supplementary Fig. 3d, it is supplementary Fig. 2d), the main text is fine.

Reviewer #3:

Remarks to the Author:

The authors have sufficiently addressed all my concerns. Unfortunately, their attempts to improve the array data failed, but the overall story is not impacted by this. Following their rebuttal, the authors could consider adding a very brief statement on the array experiments. Otherwise it would appear useless to add this data to the manuscript at all. A suggestion:

"While the [array] data were one of our first indications for a (positive) interaction [of the proteins],"

combining with their sentence starting in line 207:

"These experiments suffered from high background in the negative control, hampering straightforward interpretation (Supplementary Fig. 3a)."

POINT-BY-POINT REPLY TO REVIEWER COMMENTS (REPLIES IN BLUE)

REVIEWERS' COMMENTS

Reviewer #1 (Remarks to the Author):

The revised version of the manuscript by Borgert et al., addresses all the issues that I included in my review and provide solid data on which the author based their conclusions. I agree with the change of the title of the article since it more clearly describe the conclusions attained. Among the new data contained in the modified manuscript is the assessment of RhIR solubility and it is clearly shown that it is increased by the complex formed with PqsE and that RhIR interaction with C4-HSL is required independently of PqsE. These results explain the observations reported previously that were interpreted to be due to RhIR stability.

I consider that these results and the detailed structural characterization of RhIR coupled with C4-HSL or with mBTL, as well as of the complex formed with PqsE that requires the dimerization of this enzyme, represent a major contribution to the field of *P. aeruginosa* QS-sensing. Therefore I recommend that this manuscript is accepted for its publication in Nature Communications.

We thank the reviewer for this positive assessment of our manuscript.

Reviewer #2 (Remarks to the Author):

The authors properly responded to all the questions I raised. Although there is a simple mistake in the reply letter (new supplementary Fig. 3d, it is supplementary Fig. 2d), the main text is fine.

We thank the reviewer for this positive assessment of our manuscript and apologize for the mix-up.

Reviewer #3 (Remarks to the Author):

The authors have sufficiently addressed all my concerns. Unfortunately, their attempts to improve the array data failed, but the overall story is not impacted by this. Following their rebuttal, the authors could consider adding a very brief statement on the array experiments. Otherwise it would appear useless to add this data to the manuscript at all. A suggestion:

"While the [array] data were one of our first indications for a (positive) interaction [of the proteins],"
combining with their sentence starting in line 207:

"These experiments suffered from high background in the negative control, hampering straightforward interpretation (Supplementary Fig. 3a)."

We thank the reviewer for this excellent suggestion, which we have incorporated into the re-revised manuscript (lines 208-209).